# Efficient and Safe Molecular Assembly via Reinforcement Learning and Constraint Solving

**Stefan Pranger** [1]  **Bernhard Ramsauer** [2]  **Oliver T. Hofmann** [2]  **Bettina Könighofer** [1]

## Abstract

Scanning tunneling microscopy (STM) enables precise manipulation of individual atoms and molecules, offering a pathway to constructing nanoscale assemblies with rich quantum mechanical behavior. Despite its potential, STM-based fabrication remains limited by the inherent complexity of manipulation procedures and the extensive manual effort required. In this work, we take a substantial step toward autonomous manufacturing with STMs by introducing a novel AI-based planning framework for molecular assembly and a high-fidelity simulation environment. Our framework computes collision-free assembly plans that minimize the total distance traveled by molecules. Given an assignment of molecules to target positions, satisfiability solving is used to compute execution schedules in which each molecule has an empty corridor available when it is scheduled to move. Reinforcement learning (RL) agents then execute sequences of STM actions to manipulate molecules to their targets. We further introduce NANOASSEMBLYGYM, a high-fidelity simulation environment for molecular manipulation built on the GYMNASIUM API, allowing seamless integration with existing RL libraries and workflows. Using NANOASSEMBLYGYM, we demonstrate autonomous assembly of structures containing up to 420 molecules.

## 1. Introduction

The ability to manipulate molecules with subatomic precision offers significant opportunities for advancing both physics and chemistry (Feynman, 2011; Pitcher, 2006).

[1]Institute of Information Security, Graz University of Technology [2]Institute of Solid State Physics, Graz University of Technology, NAWI Graz. Correspondence to: Stefan Pranger <stefan.pranger@tugraz.at>.

*Proceedings of the 43rd International Conference on Machine Learning*, Seoul, South Korea. PMLR 306, 2026. Copyright 2026 by the author(s).

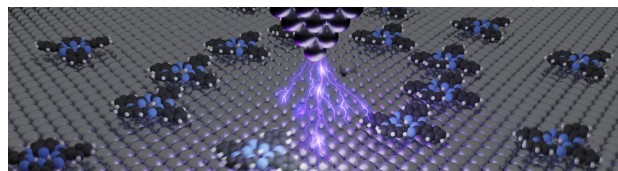

*Figure 1.* Illustration of an STM tip applying a voltage pulse to manipulate an adsorbed molecule.

Scanning tunneling microscopy (STM) enables the precise manipulation of individual atoms and molecules (Eigler & Schweizer, 1990). It has been employed to synthesize molecules inaccessible by wet chemistry (Albrecht et al., 2022) and to create logic circuits (Heinrich et al., 2002), memories (Kalff et al., 2016), and even works of art (Guinness World Records, 2013). Other applications include the arrangement of individual atoms and molecules into *quantum corrals* (Crommie et al., 1993), which enable the controlled shaping of quantum-mechanical wave functions (Stilp et al., 2021) and are considered highly promising building blocks for future quantum computing technologies (Correa et al., 2005).

STM-based molecular positioning requires the operator to perform a sequence of actions, such as precise tip placement relative to the molecule and the application of a voltage pulse (Hla, 2005), as illustrated in Fig. 1.

Although STM-based fabrication is highly promising, it is hampered by the inherent complexity of the manipulation procedure and the *extensive manual effort* it requires. Human experts may spend months constructing assemblies containing only about 100 atoms and molecules. Previous *autonomous assembly algorithms* target simple adsorbates such as atoms or CO molecules. Complex molecules enable richer assemblies but complicate manipulation by introducing an additional rotational degree of freedom. Molecular assemblies are therefore more challenging, as molecules often respond non-intuitively and stochastically to manipulation parameters, particularly when additional rotational degrees of freedom are involved (Lorente et al., 2005; Meyer et al., 2001; Morgenstern et al., 2013; Whitesides et al., 1991; Rutter, 2023). Consequently, molecular assembly involves a *high-dimensional action space* with *probabilistic responses*,

making control difficult. In practice, surface defects and malformed molecules further hinder the assembly process.

*The goal of this paper is to make a substantial step toward autonomous manufacturing with STMs by (1) proposing a novel AI-based planning framework for molecular assembly and (2) introducing high-fidelity simulation software.*

**Assembly Construction.** Our planning framework first computes a high-level *construction plan* that assigns each molecule to a target position in the assembly such that it can be moved without collisions. The algorithm proceeds iteratively: it determines an assignment of molecules to target positions that minimizes the total travel distance, and then uses satisfiability solving to construct a feasible schedule ensuring that each molecule has a free corridor during its move. If no feasible schedule is found, the molecule–target assignments causing the conflict are excluded, and the assignment step is repeated. To execute the construction plan, we train *reinforcement learning* (RL) agents to move individual molecules to their assigned target positions.

**NANOASSEMBLYGYM.** We introduce NANOASSEMBLY-GYM, a high-fidelity simulation environment for STM-based molecular assembly. Built on the GYMNASIUM API, it provides a standardized interface for training and evaluating RL agents for molecular manipulation.

**Experiments.** We evaluate our framework in NANOASSEM-BLYGYM on two types of synthetic molecules and on multiple assemblies containing up to 420 molecules. In each experiment, molecules and obstacles are randomly placed on the surface. The experiments show that high-level construction plans can be computed within a few minutes for the largest assemblies and that the trained RL agents construct the assemblies successfully in almost all trials, with success rates approaching 100% (we observed only 10 collisions when moving $> 30{,}000$ molecules).

## 1.1. Related Work

**Machine Learning in Nanophysics.** In recent years, several works studied the application of machine learning to automate construction of assemblies via STMs. Machine learning has been used primarily for the analysis and processing of STM signals (Chen et al., 2023), as well as for image classification tasks such as the automatic recognition of molecules in various states (Zhu et al., 2022; Ziatdinov et al., 2022; Krull et al., 2020), the identification of nanowires (Bai & Wu, 2021), and the detection of surface defects (Croshaw et al., 2020). ML has further been applied to the preparation of STM tips (Chen et al., 2025; Rashidi & Wolkow, 2018), to the autonomous optimization of imaging parameters such as the tip bias (Narasimha et al., 2024), and has also been used to automate the collection and categorization of STM images (Barker & Sweetman, 2025).

Autonomous manipulation of objects has so far mostly been limited to individual atoms or non-interacting molecules (Ramsauer et al., 2023; Chen et al., 2022). Ramsauer et al. train tabular Q-learning agents directly on the physical machine to move single molecules. Chen et al. train deep RL agents to move individual atoms, also directly on the real machine. Similar to our work, Rutter (Rutter, 2023) presented an approach to train deep RL agents in simulation to follow predefined paths between the start and target position.

In contrast, we train deep RL agents in simulation to manipulate molecules and propose a novel planning algorithm that computes collision-free construction plans for assembling complex molecular structures. At present, the construction of large molecular structures remains beyond current capabilities, limiting the systematic study of extended assemblies and restricting the practical use of STM-based fabrication. A realistic simulation environment constitutes a critical step toward enabling the assembly of structures comprising hundreds of molecules.

## 2. Background

**Basic Notation.** We denote by $[\![a, b]\!]$ the set of integers $i$ with $a \leq i \leq b$. Let $\mathcal{D} \in \mathbb{R}^{n \times m}$. We denote by $\mathcal{D}[i, j]$ the matrix entry in row $i$ and column $j$. For any two points $p, q \in \mathbb{R}^2$, we denote the vector between $p$ and $q$ by $\overrightarrow{pq}$ and the Euclidean distance between $p$ and $q$ as $d(p, q) = |\overrightarrow{pq}|$. By $\mathcal{B}_r(p) = \{x \in \mathbb{R}^2 \mid d(x, p) \leq r\}$, we denote the *Euclidean ball* in $\mathbb{R}^2$ around point $p$ with radius $r$. For two sets of points $A, B \subseteq \mathbb{R}^2$, we use $\oplus$ to denote the *Minkowski sum* of $A$ and $B$: $A \oplus B = \{a + b \mid a \in A, \, b \in B\}$.

**Reinforcement Learning (RL).** A *Markov Decision Process* (MDP) is a tuple $\mathcal{M} = \langle \mathcal{S}, s_0, \mathcal{A}, \mathcal{P}, \mathcal{R} \rangle$ where $\mathcal{S}$ is a finite set of states, $s_0 \in \mathcal{S}$ is the initial state, $\mathcal{A}$ is a finite set of actions, $\mathcal{P} : \mathcal{S} \times \mathcal{A} \to Dist(\mathcal{S})$ is the probabilistic transition function, and $\mathcal{R} \colon \mathcal{S} \times \mathcal{A} \times \mathcal{S} \to \mathbb{R}$ is the *reward function*. In RL (Sutton & Barto, 1998), an *agent* learns a task via interactions with an unknown *environment* modeled by an MDP $\mathcal{M}$. At each step $t$, the agent selects an action $a_t$. From the current state $s_t$, the environment then moves to state $s_{t+1}$ with probability $\mathcal{P}(s_t, a_t, s_{t+1})$, which determines the next reward $r_{t+1} = \mathcal{R}(s_t, a_t, s_{t+1})$. An RL agent seeks to learn a *policy* $\pi(a|s)$ that maximizes the expected return, expressed as the discounted cumulative reward $\mathbb{E}\left[\sum_{t=0}^{\infty} \gamma^t R(s_t)\right]$ with discount factor $\gamma$.

**Boolean Satisfiability Problem (SAT) and Satisfiability Modulo Theories (SMT).** A *Conjunctive Normal Form* (CNF) formula $\varphi$ is a conjunction of *clauses* $\mathcal{C}_i$, each a disjunction of *literals* ($x$ or $\neg x$). Typically, a CNF formula $\varphi$ is represented as the set of its clauses. Given a propositional formula $\varphi$ in CNF over a finite set of Boolean variables

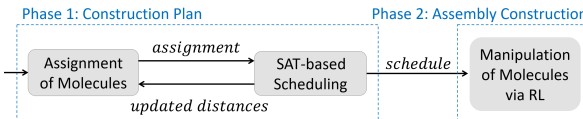

Figure 2. Construction algorithm for molecular assembly. Phase 1 *assigns* molecules to target positions and defines the placement *schedule*, while Phase 2 *executes* the placement of each molecule to its target configuration in the assembly.

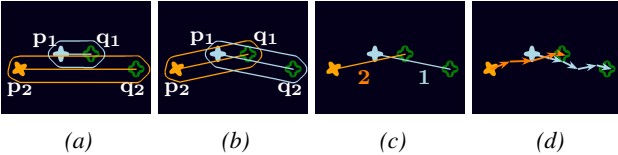

*(a)*  *(b)*  *(c)*  *(d)*

Figure 3. Example execution of the construction algorithm: (a) First matching of molecules ($p_i$) to target positions ($q_i$). (b) Second matching of molecules to target positions. (c) The computed schedule for molecule movement. (d) The movement of molecules.

$\mathcal{V} = \{x_1, \ldots, x_n\}$, SAT asks whether there exists an interpretation $\mathcal{I} : \mathcal{V} \to \{\texttt{true}, \texttt{false}\}$ such that $\mathcal{I} \models \varphi$, i.e., $\varphi$ evaluates to $\texttt{true}$ under $\mathcal{I}$. If such an interpretation $\mathcal{I}$ exists, the formula $\varphi$ is *satisfiable*; otherwise, it is *unsatisfiable*. For an unsatisfiable formula $\varphi$, an *unsatisfiable core* is a subset of clauses $\varphi' \subseteq \varphi$ such that $\varphi'$ is unsatisfiable. An unsatisfiable core $\varphi'$ is *minimal*, iff any set of clauses $\varphi'' \subset \varphi'$ is satisfiable. The framework of SMT generalizes SAT to formulas in first-order logic interpreted over background theories such as linear arithmetic, bit-vectors, or uninterpreted functions SMT then asks whether there exists an interpretation $\mathcal{I}$ that simultaneously satisfies the Boolean structure of the formula and all theory constraints.

## 3. Building Molecular Assemblies

We consider a set of molecules $\texttt{Mol} = \{m_i | m_i = \langle p_i, \alpha_i \rangle\}$, which are each defined by their current position $p_i \in \mathbb{R}^2$ and their orientation $\alpha_i \in (0°, 359°]$ on the surface plane. An *assembly* is a structured aggregate of molecules, represented as $\texttt{Ass} = \{t_j | t_j = \langle q_j, \beta_j \rangle\}$ where each target configuration $t_j$ specifies the target position $q_j$ and orientation $\beta_j$ of the molecule. Our objective is to move each molecule to its designated configuration $q_j$ in a *safe* and *efficient* manner.

### 3.1. Overview of the Construction Algorithm

The construction algorithm consists of two phases, as illustrated in Fig. 2. In Phase 1, we generate a high-level *construction plan* by assigning molecules to target positions and determining a schedule that specifies the order in which molecules are moved. In Phase 2, the *assembly construction* is executed according to the construction plan.

**Phase 1: Construction Plan (Section 3.2).** This phase uses an iterative algorithm to compute a high-level plan that *minimizes the total distance* the molecules must travel while ensuring that the plan can be *executed without collisions*. The algorithm alternates between computing an *assignment* that minimizes the total movement distance and searching for a feasible *schedule* that allows safe execution.

Moving a molecule safely from its start to target position requires an empty *corridor* connecting the two. Because molecular motion is probabilistic, this corridor must be sufficiently wide to keep the molecule within its bounds during

manipulation. A *feasible schedule* specifies an execution order where each molecule, when moved, has an empty corridor available. If no feasible schedule exists, the algorithm computes a new assignment by setting the distances between molecule-target pairs that could not be scheduled to *infinity*, thereby enforcing their reassignment in the next iteration. We formulate the problem of determining the existence of a feasible schedule as an SMT problem instance: if the formula is unsatisfiable, no feasible schedule exists. From the unsatisfiable core, we identify the conflicting assignments and adjust their distances for the next iteration.

**Phase 2: Assembly Construction (Section 3.3).** Individual molecules are moved to their target positions by executing a sequence of STM actions that consist of positioning the STM tip next to the molecule and applying a pulse at a predetermined voltage. The responses of molecules to STM actions can be modelled as a Markov decision process (MDP). We employ reinforcement learning (RL) to learn a policy that selects actions to reach the target position and orientation in as few steps as possible while keeping the molecule within its corridor with high probability.

**Example.** Fig. 3 provides an example execution of the algorithm, showing how conflicts that occur during the assembly process are resolved. In the first iteration, the computed assignment matches $p_1$ to $q_1$ and $p_2$ to $q_2$, depicted in Fig. 3a. The SMT-based scheduling algorithm is unable to compute a feasible assignment as $m_2$ can never obtain a free corridor. The second iteration produces a different matching, shown in Fig. 3b, for which the SMT-based scheduler is able to find a schedule, shown in Fig. 3c: by moving $m_1$ to $q_2$ first, it leaves a free corridor so that $m_2$ can move second. Finally, an RL agent executes the construction plan by first moving $m_1$ to $q_2$ followed by moving $m_2$ to $q_1$, thereby completing the assembly as depicted in Fig. 3d.

### 3.2. Computation of the Construction Plan

Algorithm 1 gives the pseudocode for computing the high-level plan for constructing the assembly. The algorithm iteratively computes assignments of molecules to target positions in the assembly (Lines 3-6) and then checks whether the current matching admits a feasible schedule (Lines 7-11). If not, the algorithm updates the distance matrix and

**Algorithm 1** Computation of Construction Plans

**Input**: `Mol, Ass`
**Output**: A schedule `Schedule : (Match)` $\rightarrow [\![1, n]\!]$

1: $\mathcal{D} \leftarrow$ computeDistanceMatrix(`Mol, Ass`)
2: **loop**
3:     `Match` $\leftarrow$ hungarianMatching($\mathcal{D}$)
4:     **if** no `Match` can be computed **then**
5:         **return**
6:     **end if**
7:     `Cons` $\leftarrow$ computeConstraints(`Match`)
8:     $\varphi \leftarrow$ constructSchedulingFormula(`Cons`)
9:     **if** $\varphi$ is satisfiable **then**
10:       `Schedule` $\leftarrow$ getModel($\varphi$)
11:       **return** `Schedule`
12:     **else**
13:       `Conflicts` $\leftarrow$ extractUnsatCore($\varphi$)
14:       $\mathcal{D} \leftarrow$ updateDistanceMatrix($\mathcal{D}$, `Conflicts`)
15:     **end if**
16: **end loop**

proceeds to the next iteration (Lines 12-15). The individual steps of the algorithm are discussed in detail below.

**Compute Assignment of Molecules.** A matching `Match` $= \{(p_i, q_j) \mid \langle p_i, \alpha_i \rangle \in$ `Mol`$, \langle q_j, \beta_j \rangle \in$ `Ass`$\}$ assigns each molecule position $p_i$ to a target position $q_j$ in the assembly. We note here that the orientation of the molecules does not influence the computation of the construction plan.

In Line 1 we first compute the distance matrix $\mathcal{D} \in \mathbb{R}^{n \times n}$ that captures the Euclidean distances $d(p_i, q_j)$ between all molecule positions $p_i$ and possible target positions $q_j$. Next, we use the *Hungarian algorithm* (Kuhn, 1955) to compute a matching `Match` such that the sum of the Euclidean distances $\sum d(p_i, q_j)$ is minimized (Line 3). We denote the assignment of $p_i$ to target position $q_j$ as $a_{ij} = (p_i, q_j) \in$ `Match`. If the Hungarian algorithm is unable to find an assignment, the algorithm aborts (Line 4-6).

**Feasible Schedule.** Given a matching `Match`, the next step is to find a *feasible schedule*. A schedule defines the order in which the assignments $a_{ij} \in$ `Match` of molecules to target positions will be executed. A *feasible schedule* ensures for each assignment $a_{ij}$ that, at its turn, $m_i$ has an empty *corridor* available. We define a corridor between $p_i$ and $q_j$ as any convex polygon that is at least wide enough to contain the smallest enclosing ball of the molecule at both positions. Formally, let $r$ be the smallest radius such that $\mathcal{B}_r(p_i)$ fully encloses $m_i$ at $p_i$. A corridor `Corr`$(p_i, q_j)$ between $p_i$ and $q_j$ is any convex polygon that includes $\mathcal{B}_r(p_i)$ and $\mathcal{B}_r(q_j)$. We refer to Appendix E for precise geometric definitions.

**Scheduling Constraints.** A feasible schedule defines an ordering in which every molecule moves through an empty corridor at its scheduled time. Since the start or target posi-

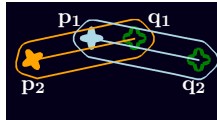
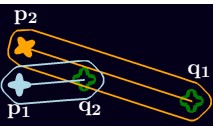

*(a)* Start precedence constraint.    *(b)* Target precedence constraint.

*Figure 4.* Examples of the two types of precedence constraints.

tion of a molecule may be contained within the corridor of another molecule, this induces constraints on the execution order of assignments. We write $a_{ij} < a_{kl}$ if the assignment $a_{ij}$ needs to be executed *before* $a_{kl}$. We distinguish between two types of constraints:

- *Start-precedence constraints:* An assignment $a_{ij}$ needs to be executed *before* $a_{kl}$ if the ball around the *start position* $p_i$ lies within the corridor between $p_k$ and $q_l$. Formally, $a_{ij} < a_{kl}$ if $\mathcal{B}_r(p_i) \cap$ `Corr`$(p_k, q_l) \neq \emptyset$.
- *Target-precedence constraints:* An assignment $a_{ij}$ needs to be executed *after* $a_{kl}$ if the ball around the *target position* $q_j$ lies within the corridor between $p_k$ and $q_l$. Formally, $a_{kl} < a_{ij}$ if $\mathcal{B}_r(q_j) \cap$ `Corr`$(p_k, q_l) \neq \emptyset$.

**Example.** Fig. 4a depicts a start precedence constraint: assignment $a_{12} = (p_1, q_2)$ needs to be executed before $a_{21} = (p_2, q_1)$. Fig. 4b shows a target precedence constraint: $a_{12} = (p_1, q_2)$ can only be executed after $a_{21} = (p_2, q_1)$.

**SMT-based Schedule Computation.** The next step in the algorithm is to collect the set of all precedence constraints `Cons` $= \{(a_{ij}, a_{kl}) \mid a_{ij} < a_{kl}\}$ (Line 7). From `Cons`, we construct an SMT formula $\varphi$ such that every satisfying assignment of $\varphi$ corresponds to a feasible schedule.

In constructing $\varphi$, each assignment $a_{ij}$ is associated with an integer $s_{ij} \in \mathbb{N}$ that represents its position in the schedule. Given $n$ molecules to be scheduled, $\varphi$ requires that each variable $s_{ij}$ is assigned a unique value in $[\![1, n]\!]$, and that the precedence constraints are respected: if $a_{ij} < a_{kl}$, then the value assigned to $s_{ij}$ must be smaller than the value assigned to $s_{kl}$. This gives rise to the following SMT formula (Line 8).

$$\varphi = \bigwedge_{\substack{a_{ij}, a_{kl} \in \text{Match}, \\ a_{ij} \neq a_{kl}}} s_{ij} \neq s_{kl} \ \wedge \bigwedge_{(a_{ij}, a_{kl}) \in \text{Cons}} s_{ij} < s_{kl}.$$

$\varphi$ is checked for satisfiability in Line 9 using standard tools for solving SMT formulas (e.g., SMT Solver Z3 (De Moura & Bjørner, 2008)). If $\varphi$ is satisfiable, the values assigned to each $s_{ij}$ define a feasible schedule `Schedule : Match` $\rightarrow [\![1, n]\!]$. (Line 10-11).

**Conflict Extraction and Distance Adaptation.** If $\varphi$ is unsatisfiable, then no feasible schedule exists for the current assignment `Match`. In this case, the SMT solver returns a minimal unsatisfiable core `Conflicts` $= \{(a_{ij}, a_{jk}), \dots\} \subseteq$

Cons that contains the constraints that could not be satisfied (Line 13). To obtain a new matching in the next iteration of the algorithm, we set certain distances in the distance matrix $\mathcal{D}$ to $\infty$ according to Conflicts: for each $(a_{ij}, a_{kl}) \in$ Conflicts, we set $\mathcal{D}[i, j] = \infty$ (Line 14).

**Example.** In Fig. 3, consider the first assignment (left panel) with Match $= \{a_{11}, a_{22}\}$, where $a_{11} = \langle(p_1, \alpha_1), (q_1, \beta_1)\rangle$ and $a_{22} = \langle(p_2, \alpha_2), (q_2, \beta_2)\rangle$. The resulting formula $\varphi$ is unsatisfiable, because the precedence constraints $s_{11} < s_{22}$ and $s_{22} < s_{11}$ cannot be satisfied simultaneously. Thus, the algorithm will either return the unsatisfiable core Conflicts $= \{(a_{11}, a_{22})\}$ or Conflicts $= \{(a_{22}, a_{11})\}$.

### 3.3. Assembly Construction

To construct the assembly, we follow the construction plan defined by the assignments in Match and the feasible schedule Schedule. In the order specified by Schedule, we execute each assignment $a_{ij} = \langle(p_i, \alpha_i), (q_j, \beta_j)\rangle$ by moving molecule $m_i$ to its target configuration $(q_j, \beta_j)$.

The movement of a molecule to its target configuration in the assembly is realized by applying a sequence of STM actions. We use RL to learn an optimal policy that moves the molecule to its target position and orientation with a minimal number of actions while remaining within the empty corridor. In the following, we define the RL problem for moving a molecule $m = \langle p, \alpha \rangle$ to a configuration $\langle q, \beta \rangle$.

**Environment modeling.** We model the probabilistic movements of the molecule as an MDP $\mathcal{M} = (\mathcal{S}, s_0, \mathcal{A}, \mathcal{P}, \mathcal{R})$. A state $s = (p, \alpha, q, \beta) \in \mathcal{S}$ represents the current position $p$ and orientation $\alpha$ of the molecule, the target position $q$, and the target orientation $\beta$. The action space $\mathcal{A}$ is centered at the molecule's position $p$ and aligned with its orientation $\alpha$. An action $a = (x, y) \in \mathcal{A}$ represents placing the tip at position $(x, y)$ relative to $p$ and rotated by $\alpha$, and applying a voltage pulse. $\mathcal{A}$ is discretized by the *lattice constant $L$* of the surface, corresponding to the nearest-neighbor spacing of surface atoms. The probabilistic transition function $\mathcal{P}$ updates the position and orientation of the molecule. After applying an action $(x, y)$, the position $p$ changes to a new position $p'$ according to a multivariate Gaussian distribution $\mathcal{N}(\mu_{x,y}, \sigma_{x,y}^2)$. Changes in the orientation $\alpha$ are modeled by a discrete probability distribution $A_{x,y}$ over a finite set $\rho$ of possible rotations. The probabilistic updates are assumed to be independent. The complete update function is:

$$\mathcal{P}(s, (x, y)) = (p + \mathcal{N}(\mu_{x,y}, \sigma_{x,y}^2), \alpha + A_{x,y}(\rho), q, \beta).$$

**Reward function.** The reward function $\mathcal{R}$ is designed to guide the molecule's movement toward the target configuration: the greater the progress toward the target, the higher the reward. Actions that move the molecule away from the goal or to leave the corridor incur a negative reward. The reward $\mathcal{R}$ is the sum of three components: (i) $\mathcal{R}_{move}$, measuring progress towards the target position; (ii) $\mathcal{R}_{corr}$, capturing violations of the corridor constraint; and (iii) $\mathcal{R}_{rot}$, measuring progress in orientation toward the target orientation. The function $\mathcal{R}_{move}$ is defined as follows:

$$\mathcal{R}_{move}(s, a, s') = \tanh\left(\frac{d(p, q) - d(p', q) - L}{k \cdot max_{move}}\right),$$

with $s = (p, \alpha, q, \beta)$ and $s' = (p', \alpha', q, \beta)$. To penalize small movements, the lattice constant $L$ is subtracted from the translation. The distance is normalized by $k \cdot max_{move} = k \cdot \max_{(x,y)}\{\mu_{x,y}\}$. The value $max_{move}$ is determined as the maximum average (mean) movement encountered for any allowed action. The constant $k = 0.5$ in the denominator sharpens the transition of the tanh, yielding larger gradients in the critical region. $\mathcal{R}_{corr}$ captures corridor violations and is defined as:

$$\mathcal{R}_{corr}(s, a, s') = \begin{cases} 0 & \text{if } \mathcal{B}_r(p) \subseteq \text{Corr}(p, q) \\ -1.5 & \text{else,} \end{cases}$$

inducing a penalty whenever the molecule moves outside its corridor. Finally, $\mathcal{R}_{rot}$ rewards rotations towards the target orientation. Let $d_{\text{circ}}(\beta, \alpha) = \left|\arg\left(e^{i(\beta - \alpha)}\right)\right|$, be the distance between angles $\beta$ and $\alpha$, with $\arg(\cdot)$ denoting the principal argument in $(-\pi, \pi]$. $\mathcal{R}_{rot}$ is defined as:

$$\mathcal{R}_{rot}(s, a, s') = \begin{cases} 0 & \text{if } d(p, q) \geq \varepsilon_{\text{rot}}, \\ -0.5 & \text{else if } d_{\text{circ}}(\beta, \alpha') > d_{\text{circ}}(\beta, \alpha), \\ 0.5 & \text{else if } d_{\text{circ}}(\beta, \alpha') < d_{\text{circ}}(\beta, \alpha), \\ 0 & \text{else.} \end{cases}$$

Whenever the molecule is $\varepsilon_{rot}$-close to the target position, rotating towards the target orientation $\beta$ is rewarded, while rotating away is penalized.

### 3.4. Constraint-Based Partitioning

In Sec. 3.2, we solved the problem of finding a feasible schedule as a global constraint-solving task. However, for large assemblies with hundreds of molecules, solving the resulting problem can become costly. We present an extension of Algorithm 1 that introduces a partitioning of the problem space. Intuitively, assignments that are not spatially connected do not influence each other's order and can therefore be scheduled independently. This leads to smaller problem instances and fewer iterations of the overall algorithm, since unsatisfiable constraints are identified per subproblem. The pseudocode of the extended algorithm, including partitioning, is given in Appenix A.

In the extension, we define *classes of precedence-related assignments* PRClass, where each class includes all as-

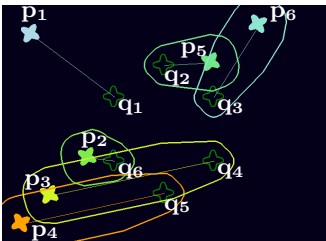

*Figure 5.* A matching that is partitioned into two sub-problems.

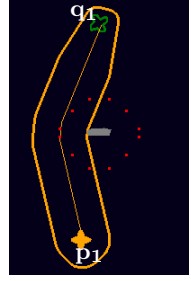 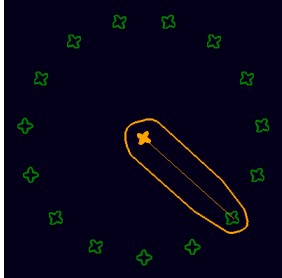

*(a)* Obstacles avoidance: Red points depict waypoints used to compute circumventing paths.

*(b)* Environment for RL training. At each episode, a random target position is chosen.

*Figure 6.* Circumventing corridors and RL training environment.

signments that are related via precedence constraints. Concretely, if an assignment $a_{ij}$ is in a class $\texttt{PRClass}_k$, then for all $(a_{ij} \not z a_{kl}) \in \texttt{Cons}$ it holds that $a_{kl} \in \texttt{PRClass}_k$, where $a_{ij} \not z a_{kl}$ is either $a_{kl} < a_{ij}$ or $a_{ij} < a_{kl}$. The feasible schedule $\mathcal{S}$ is then constructed incrementally: For each $\texttt{PRClass}_k$, an ordering of $a_{ij} \in \texttt{PRClass}_k$ is computed individually by solving the SMT problem as described in Sec. 3.2. If the problem is satisfiable, the solution yields a feasible schedule for all $a_{ij} \in \texttt{PRClass}_k$. For each unsatisfiable class $\texttt{PRClass}_k$, we collect the corresponding unsatisfiable constraints and adapt $\mathcal{D}$ as outlined in Sec. 3.2.

**Example.** Consider the matching depicted in Fig. 5. The assignments can be partitioned into two classes: $\texttt{PRClass}_1 = \{a_{26}, a_{34}, a_{45}\}$ and $\texttt{PRClass}_2 = \{a_{52}, a_{63}\}$. The unconstrained assignment $a_{11}$ can be scheduled arbitrarily.

### 3.5. Construction Plan with Obstacle Avoidance

In this section, we present an extension of Algorithm 1 that takes obstacles on the surface into account. To that end, we adapt distances in $\mathcal{D}$ such that a distance $\mathcal{D}[i,j]$ corresponds to the length of the *shortest circumventing path* that avoids any obstacles between $p_i$ and $q_j$. These paths are computed by introducing *waypoints* around obstacles. Connecting the start and target positions via these waypoints yields a *chain of corridors* through which the molecule can circumvent the obstacles. The pseudocode of the extended algorithm, including obstacle avoidance, is given in Appendix A.

**Example.** Fig. 6a shows the shortest path and circumventing corridor between the molecule's position $p_1$ and its target position $q_1$ that avoids the gray obstacle.

Let $\texttt{Obst} = \{o_i \mid o_i \subset \mathbb{R}^2\}$ be a set of convex polygonal obstacles that are randomly distributed on the surface. For a corridor of width $w$, the vertices $\texttt{WayP}(o_i, w)$ define the smallest polygon containing all points whose distance to $o_i$ is at most $\frac{w}{2}$. A corridor from $p$ to $q$ does not intersect $o_i$ if the centerline $\overrightarrow{pq}$ of the corridor does not intersect the interior $\texttt{Int}(\texttt{WayP}(o_i, w))$ of the polygon defined by $\texttt{WayP}(o_i, w)$, i.e. $\overrightarrow{pq} \cap \texttt{Int}(\texttt{WayP}(o_i, w)) = \emptyset$. The vertices $\texttt{WayP}(o_i, w)$ are then used to compute *shortest circumventing paths*. A path $\tau = p, v_1, \ldots, v_m, q$ is the shortest path from $p$ to $q$ via the waypoints $v_1, \ldots, v_m \in \texttt{WayP}(o_i, w)$ such that $\overrightarrow{p, v_1}$, $\overrightarrow{v_m, q}$, and $\overrightarrow{v_k v_{k+1}}$ do not in-

tersect any $\texttt{Int}(\texttt{WayP}(o_i, w))$, for every $v_k, v_{k+1} \in \tau$. We obtain the shortest circumventing path between $p$ and $q$ by computing a shortest path on a weighted graph $G = (V, E, w)$ using standard algorithms. $G$ is defined via:

- $V = \{p | \langle p, \alpha \rangle \in \texttt{Mol}\} \cup \{q | \langle q, \beta \rangle \in \texttt{Ass}\} \cup \bigcup_{o_i \in \texttt{Obst}} \texttt{WayP}(o_i, w)$ consisting of all molecule positions, target positions, and waypoints generated for the obstacles $o_i \in \texttt{Obst}$,
- the edge set $E = \{(e, f) \mid e, f \in V, e \neq f\}$, and
- $w : E \to \mathbb{R}$ that assigns the distance $d(e, f)$, or infinity if $(e, f)$ intersects the interior of some $\texttt{WayP}(o_i, w)$:

$$w(e, f) = \begin{cases} d(e, f) & \text{if } \forall o_i \ \overrightarrow{ef} \cap \texttt{Int}(\texttt{WayP}(o_i, w)) = \emptyset \\ \infty & \text{else.} \end{cases}$$

In the extended algorithm, the distance between $p_i$ and $q_j$ is computed based on the shortest circumventing path $\tau = p_i, v_1, \ldots, v_m, q_j$. Therefore, we set

$$\mathcal{D}[i,j] = d(p_i, v_1) + \sum_{k=1}^{m-1} d(v_k, v_{k+1}) + d(v_m, q_j).$$

and the chain of corridors $\texttt{Corr}(p_i, v_1), \ldots, \texttt{Corr}(v_m, q_j)$ is used to compute precedence constraints.

## 4. Experimental Evaluation

We implemented our complete framework for building assemblies in the new software library NANOASSEMBLY-GYM, which offers an integrated environment for learning, evaluation, and execution. The core of the library is a simulation environment for STM manipulation based on user-provided measurements of molecule or atom responses to STM actions. NANOASSEMBLYGYM is built on the GYM-NASIUM API standard (Brockman et al., 2016), enabling RL training using state-of-the-art algorithms. Figures 3 - 9 show screenshots from the simulation. To construct an assembly in NANOASSEMBLYGYM using our planning framework, a user has to provide the following inputs: (1) A distribution

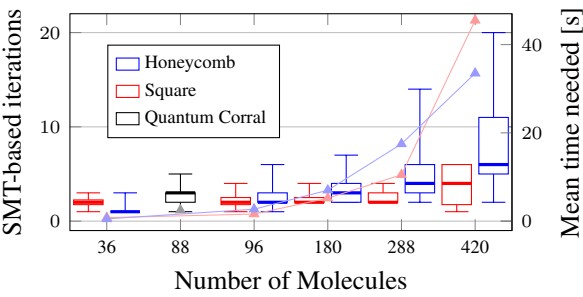

*(a)* Results for assembly construction without obstacles.

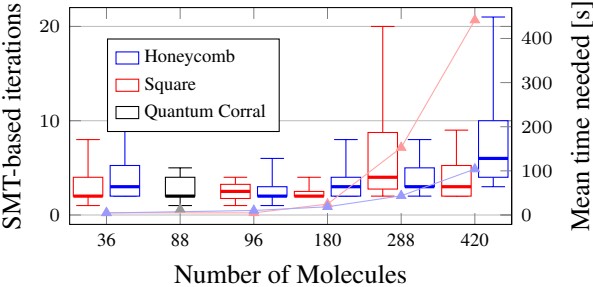

*(b)* Results for the assembly construction with obstacles.

*Figure 7.* The number of iterations (boxplots, left axis) and mean computation time for a feasible schedule (right axis) for the individual assemblies with $w = 6$ nm. Boxplots show the median (thick line), the $25^{th}$–$75^{th}$ percentiles (box), and the minimum and maximum.

for the initial positions of the molecules and obstacles on the surface, (2) the set of target configurations defining the assembly to be built: $\mathtt{Ass} = \{t_j \mid t_j = \langle q_j, \beta_j \rangle\}$, and (3) the responses of the molecules or atoms to STM manipulations, given by $\mu_{x,y}$, $\sigma^2_{x,y}$, and $A_{x,y}(\rho)$ for $(x, y) \in \mathcal{A}$.

We evaluate three assembly types using three types of molecules with distinct manipulation behavior. Sec. 4.1 describes the setup, molecules, and assemblies. Construction planning results are in Sec. 4.2, RL training in Sec. 4.3, and autonomous assembly results in Sec. 4.4. Supplementary material includes an appendix, code, and videos.

### 4.1. Experimental Setting

For all experiments, we assume a surface with lattice constant $L = 0.3$ nm. In the following, we briefly describe the behaviour of the two types of molecules used for evaluation and define the assemblies. Details of the molecular motion are provided in Appendix B.1, and illustrations of the assemblies are given in Appendix B.2.

**Circular Molecule.** The first molecule is modeled as a circular disk with radius of $1.1$ nm. Manipulating this molecule pushes it into the opposite direction of the tip position with the largest movements $max_{move} = 1$ nm expected when manipulated at a radius close to $0.9$ nm. Manipulations that are closer or further away from the center induce smaller

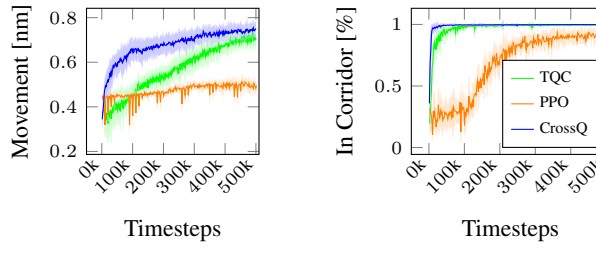

*(a)* Average movement in nm.     *(b)* Corridor adherence.

*Figure 8.* Training results for the cross-shaped averaged over 10 runs. The mean across runs is shown as a thick line, and the standard deviation as a shaded region.

movements. The covariance $\sigma_{x,y} = 0.025I$ is identical for all $(x, y) \in \mathcal{A}$ and induces displacements from $\mu_{x,y}$ by up to $L$ with $95\%$ probability. Due to its circular symmetry, this molecule does not exhibit rotational motion.

**Cross-shaped Molecules.** We model two cross-shaped molecules based on *phthalocyanine*, a well-known compound frequently studied using STMs (Neél et al., 2016). These molecules, as illustrated in Fig. 9, can be enclosed within a circle of radius $1.1$, nm. They exhibit large movements, with $max_{move} = 1.1$ nm, when manipulated on the arms, and only small movements when manipulated elsewhere. The two molecules differ in the choice of $\sigma_{x,y}$: for the first $\sigma_{x,y}$ is chosen identical to that of the circular molecule and for the second we set $\sigma_{x,y} = 0.1I$, inducing displacements from $\mu_{x,y}$ by up to $2.5L$ with $95\%$ probability. Applying pulses to the left/right of a molecular arm induces clockwise/counterclockwise rotations of $30°$, respectively, each with probability of $15\%$. The molecule can adopt orientations $\rho \in \{0°, 30°, 60°\}$.

**Assemblies.** We study three types of assemblies: (1) A quantum corral with $88$ molecules, in the geometry taken from (Correa et al., 2005), (2) *Squares* of size $n \in \{36, 96, 180, 288, 420\}$, as in (Ton et al., 2025), and (3) *Honeycomb structures* consisting of groups of molecules arranged in a lattice inspired by the lattice of graphene.

### 4.2. Computation of Construction Plans

For each assembly, we computed construction plans using *corridor widths* $w \in \{4, 5, 6\}$ nm for 20 initial distributions without obstacles and 20 with obstacles covering $0.3\%$ of the surface area. The results for $w = 6$ nm shown in Fig. 7 report the number of iterations and the mean time required to compute the construction plans without and with obstacles. The results for $w \in \{4, 5\}$ nm are given in Appendix C. For all assemblies and initial distributions, our approach consistently computes valid construction plans. Both the number of iterations and the computation time increase with assembly size and structural complexity, but this increase is moderate, indicating that the method scales well.

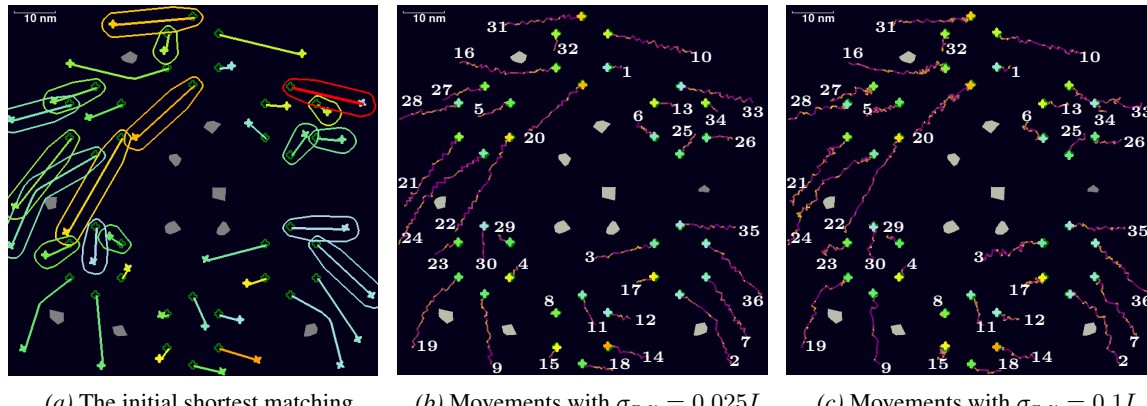

*(a)* The initial shortest matching.   *(b)* Movements with $\sigma_{x,y} = 0.025I$.   *(c)* Movements with $\sigma_{x,y} = 0.1I$.

*Figure 9.* Example executions of our approach: (a): The initial matching with corridors highlighting the constrained assignments. The assignment in red cannot be scheduled. The movement of cross-shaped molecules with (b) $\sigma_{x,y} = 0.025I$ and (c) $\sigma_{x,y} = 0.1I$.

**Structural complexity.** We compare square and honeycomb assemblies. Square assemblies induce simpler precedence constraints that can typically be resolved in few iterations, whereas the denser honeycomb layouts induce more complex constraints and require more iterations for the same number of molecules.

**Time for computing circumventing corridors.** Comparing mean computation times with and without obstacles reveals a substantial increase under the presence of obstacles, despite only a minor increase in the number of iterations. This indicates that computing circumventing corridors dominates the overall computation time. A breakdown of computation time confirms this: for honeycomb assemblies, circumventing corridor computation accounts for 55% of the total time, while SMT solving and distance matrix computation each account for 20%, with assignment computation taking the remaining 5%. For square assemblies, circumventing corridor computation is even more dominant at 75%, with distance matrix computation at 20%, and SMT solving requiring only 1% of the total time.

### 4.3. Training RL Agents

We trained RL policies to move all types of molecules to their target positions within corridor widths $w \in \{4, 5, 6\}$ nm. The action space for all molecules is given by $R = [-1.5\,\text{nm}, 1.5\,\text{nm}]^2$, discretized with step size $L$. In each episode, $q_j$ and $\beta_j$ were randomly chosen to encourage robust learning. Observations and rewards were defined as described in Sec. 3.3. We compared three RL algorithms: PPO (Schulman et al., 2017), TQC (Kuznetsov et al., 2020), and CrossQ (Bhatt et al., 2024). We provide all technical details and results in Appendix D.

**Circular Molecule.** We observed similar results across all corridor widths. All algorithms learned a policy that reliably moved the molecule to its target position within the corridor in fewer than 200k training steps. TQC converged to policies with a slightly higher average molecular movement, reaching 0.83 nm per step, compared to 0.80 nm for CrossQ and PPO. After convergence, all trained policies stay within corridors of any width 99.7% of the time.

**Cross-shaped Molecule with $\sigma_{x,y} = 0.025I$.** For the molecule with lower variance, we again observed similar results for the corridor widths $w \in \{4, 5, 6\}$ nm. Fig. 8 presents the training results for $w = 6$ nm. This time, CrossQ and TQC outperform PPO. Within 500k training steps, both CrossQ and TQC learned policies that move the molecule effectively within the corridor to its target position, with CrossQ converging more quickly. The resulting average movement is approximately 0.77 nm for CrossQ and 0.73 nm for TQC, whereas PPO converges at about 0.49 nm. After convergence, policies trained with CrossQ or TQC remain within corridors of any width 99.7% of the time.

**Cross-shaped Molecule with $\sigma_{x,y} = 0.1I$.** Across all corridor widths, CrossQ outperforms both TQC and PPO, converging to a higher average movement than TQC, while PPO fails to keep the molecule within the corridor. However, the narrower the corridor, the more often the molecule is moved outside of the corridor. In particular, for $w = 4$ nm, CrossQ and TQC maintain corridor confinement 95% of the time. For corridor widths of 5 and 6 nm, this increases to 98.3% and 98.7%, respectively. For all corridor widths, CrossQ consistently outperforms TQC in terms of average movement, exceeding it by at least 0.1 nm and attaining 0.8 nm versus 0.66 nm at $w = 6$ nm.

### 4.4. Assembly Construction

We assessed the full framework using a TQC policy for the circular molecule and CrossQ policies for the cross-shaped molecules. Each of the 11 assemblies was constructed five times with randomized placements of molecules and obstacles. A molecule was considered in its target configuration

if the distance between its center and the target position was less than $L$, with correct orientation also required for cross-shaped molecules. Across all experiments, $10\,640$ molecules were moved per molecule type. All assemblies were built using a corridor width of $w = 6\,\text{nm}$.

**Circular Molecule.** The circular molecule remained within the corridor for $99.7\%$ of manipulations, with a mean movement of $0.83\,\text{nm}$. Using this molecule, we successfully built all 55 assemblies without collisions.

**Cross-shaped Molecule with $\sigma_{x,y} = 0.025I$.** This molecule remained within the corridor for $99.8\%$ of manipulations, with a mean movement of $0.73\,\text{nm}$. Across all 55 runs, only one assembly was not completed correctly, with 2 molecules having collided.

**Cross-shaped Molecule with $\sigma_{x,y} = 0.1I$.** Across all manipulations, a corridor confinement rate of $97.3\%$ was achieved, with a mean movement of $0.63\,\text{nm}$. Five of the 55 assemblies were not completed successfully, with a total of eight molecule collisions observed.

## 5. Sim-to-Real Discussion

Transitioning our framework to a real STM setup requires addressing the following challenges, which we identify as directions for future work.

**Changes to the STM Tip.** The biggest challenge lies in the fact that the tip may change during operation. In cases where the tip absorbs an atom or molecule, the response to manipulations changes significantly, such that controlled manipulation is no longer possible even with robust RL policies. In order to ensure continuous execution, we will incorporate monitoring mechanisms to detect tip changes, and develop recovery mechanisms based on RL-based tip conditioning (Chen et al., 2025).

**Model Inaccuracies.** The responses of molecules to manipulation are inferred from real-world measurements of the molecules' reactions. Given the vast parameter space, the inferred distributions may be imprecise due to the complex, probabilistic nature of molecular motion and the limited number of available measurements. In future work, we will explore algorithms for training RL policies that ensure safe manipulation given this epistemic uncertainty.

**Failure Recovery.** In the rare event that a molecule leaves its assigned corridor, the current framework continues execution without online re-planning. For real-world STM control, where a collision can damage the tip or contaminate the surface, a failsafe mechanism is necessary. We will extend NANOASSEMBLYGYM to automatically trigger a failsafe controller upon corridor violations, enforcing that the molecule is guided back inside the corridor before execution resumes.

Not affected by the sim-to-real gap is the planning layer: the computation of distance-optimal assignments, collision-free schedules, and circumventing corridors are purely geometric and algorithmic, and transfer directly to real-world settings. NANOASSEMBLYGYM further performs exact computations of molecule position and orientation at the nanometer scale using a configurable lattice constant, without introducing abstractions or discretisation.

## 6. Conclusion & Future Work

This paper advances autonomous STM manufacturing by introducing a novel planning framework for nanostructure assembly and NANOASSEMBLYGYM, a high-fidelity simulation environment. The framework computes distance-optimal assignments and collision-free schedules, executed by RL agents trained to manipulate molecules to their target configurations efficiently and safely. In future work, we plan to close the sim-to-real gap as outlined in Section 5, and extend the RL action space to include lateral tip movement and control over bias voltage, pulse duration, and tip height, unlocking more precise and versatile STM manipulation.

We gratefully acknowledge funding from the Austrian Science Fund (FWF) through grants AI-VAMOS (10.55776/FG3400), BilAI (10.55776/COE12), and ESPRIT (10.55776/ESP1120124). We thank Filip Cano Cordoba for stimulating discussions, proof-reading of the manuscript and for sharing his experiences on related topics.

## Impact Statement

This paper presents work whose goal is to advance the field of surface science and nanoscale fabrication by enabling more autonomous, efficient, and reliable molecular assembly using scanning tunneling microscopy. There are no potential societal consequences of our work that we believe require specific discussion beyond its contribution to fundamental research and enabling technologies.

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

---

**Algorithm 2** Computation of Construction Plans

---

**Input**: `Mol`, `Ass`
**Output**: A schedule $\mathtt{Schedule} : (\mathtt{Match}) \to [\![1, n]\!]$

 1: $\mathtt{SP}, \mathtt{CC} \leftarrow$ computeCircumventingCorridors($\mathtt{Mol}, \mathtt{Ass}, \mathtt{Obst}$)
 2: $\mathcal{D} \leftarrow$ computeDistanceMatrix($\mathtt{Mol}, \mathtt{Ass}, \mathtt{SP}$)
 3: **loop**
 4:   $\mathtt{Match} \leftarrow$ hungarianMatching($\mathcal{D}$)
 5:   **if** no $\mathtt{Match}$ can be computed **then**
 6:     **return**
 7:   **end if**
 8:   $\mathtt{Cons} \leftarrow$ computeConstraints($\mathtt{Match}, \mathtt{CC}$)
 9:   $\mathtt{CAss} \leftarrow$ assignmentsWithConstraints($\mathtt{Match}, \mathtt{Cons}$)
10:   $\{\mathtt{PRClass}_1, \dots\} \leftarrow$ partition($\mathtt{CAss}$)
11:   $\mathtt{Conflicts} \leftarrow \emptyset$
12:   $u \leftarrow |\mathtt{Match} \setminus \mathtt{CAss}|$
13:   $\mathtt{Schedule} \leftarrow \mathtt{Match} \setminus \mathtt{CAss} \to [\![1, u]\!]$
14:   **for all** $\mathtt{PRClass}_k \in \{\mathtt{PRClass}_1, \dots\}$ **do**
15:     $\varphi_k \leftarrow$ constructSchedulingFormula($\mathtt{PRClass}_k$)
16:     **if** $\varphi_k$ is satisfiable **then**
17:       $\mathtt{Schedule} \mathrel{+}=$ getModel($\varphi_k$)
18:     **else**
19:       $\mathtt{Conflicts} \mathrel{+}=$ extractUnsatCore($\varphi_k$)
20:     **end if**
21:   **end for**
22:   **if** $\mathtt{Conflicts}$ is empty **then**
23:     **return** $\mathtt{Schedule}$
24:   **else**
25:     $\mathcal{D} \leftarrow$ updateDistanceMatrix($\mathcal{D}, \mathtt{Conflicts}$)
26:   **end if**
27: **end loop**

---

## A. Complete Algorithm

Sec. 3.2 outlines the basic algorithm to compute construction plans without extensions. In Sec. 3.4, we discuss how the scalability of our approach can be increased by partitioning all assignments into classes of precedence-related assignments. In Sec. 3.5, we introduce an extension to our algorithm to account for obstacles on the surface. In the following, we provide the complete algorithm combining both extensions. We first show how obstacle avoidance is applied as a preprocessing step, and then outline the changes needed to compute a feasible schedule based on the partitioned classes of precedence-related assignments. The resulting extended algorithm is summarized in Algorithm 2.

**Computing Shortest Circumventing Corridors.** Computing corridors that circumvent immovable obstacles on the surface is performed once for each assembly construction. In Sec. 3.5, we discuss the computation of shortest circumventing paths and corresponding chains of corridors that allow molecules to be moved safely around obstacles.

Before computing the distance matrix $\mathcal{D}$ in Line 2, the extended algorithm first computes the shortest circumventing paths $\mathtt{SP}$ and the associated circumventing corridors $\mathtt{CC}$ in Line 1. The computation of $\mathcal{D}$ in Line 2 then uses the total length of the shortest path between any molecule position $p_i$ and target position $q_j$ whenever the direct corridor $\mathtt{Corr}(p_i, q_j)$ overlaps with an obstacle. Accordingly, the computation of precedence constraints in Line 8 is extended to make use of the precomputed circumventing corridors $\mathtt{CC}$.

**Constraint-Based Partitioning.** Partitioning the set of assignments changes how conflicts or a feasible schedule is computed. Based on the outcome of each individual subproblem in the partition, either the set of conflicts or the current candidate schedule is incrementally extended.

In each iteration, we first compute a matching $\mathtt{Match}$ in Line 4 and, based on $\mathtt{Match}$, the corresponding precedence

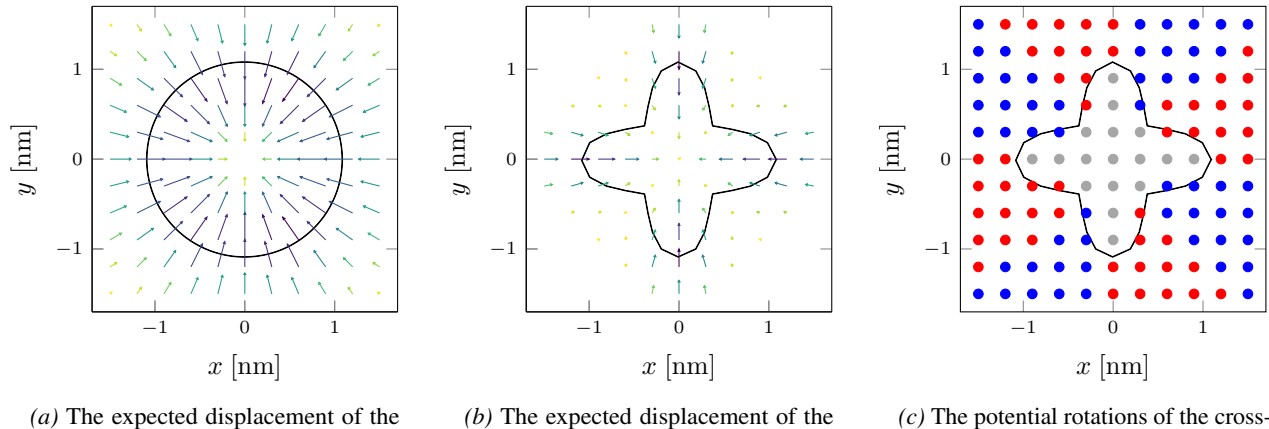

*(a)* The expected displacement of the circular molecule under manipulation.

*(b)* The expected displacement of the cross-shaped molecules.

*(c)* The potential rotations of the cross-shaped molecules under manipulation.

*Figure 10.* Expected translational and rotational behaviour of the two types of molecules under manipulation.

constraints `Cons` in Line 8. In Line 9, we compute the set of constrained assignments `CAss`, which is then partitioned into individual classes `PRClass`$_k$ in Line 10, as described in Sec. 3.4. Before determining an order of assignments within each class `PRClass`$_k$, the conflict set `Conflicts` is initialized as the empty set in Line 11, and the schedule `Schedule` is initialized with all unconstrained assignments in `Match` \ `CAss` in Line 13. Since unconstrained assignments can be scheduled arbitrarily, they are randomly assigned an order in $[\![1, u]\!]$, where $u = |\text{Match} \setminus \text{CAss}|$ denotes the number of unconstrained assignments.

The algorithm then constructs the SMT formula $\varphi_k$ for each class `PRClass`$_k$ and checks it for satisfiability in Line 15 and Line 16, respectively. If $\varphi_k$ is satisfiable, the current candidate schedule `Schedule` is extended using the values assigned to each $s_{ij}$ in the corresponding model. Otherwise, the unsatisfiable core of $\varphi_k$ is appended to the conflict set `Conflicts`.

If all $\varphi_k$ are satisfiable, i.e. `Conflicts` is empty, `Schedule` is a feasible schedule, and the algorithm returns it in Line 23. If at least one $\varphi_k$ is unsatisfiable the distance matrix $\mathcal{D}$ is updated in Line 25 and a new matching `Match` is computed in the next iteration.

## B. Details for Experimental Evaluation

### B.1. Expected Behaviour of the Molecules

Fig. 10 illustrates the expected behavior of the molecules under manipulation. Each panel shows the action space of influence $R = [-1.5\,\text{nm}, 1.5\,\text{nm}]^2$ centered on the molecule, where arrows in Fig. 10a and 10b show the mean movement $\mu_{x,y}$, indicating the direction and distance of the expected movement Red, blue and gray markers in Fig. 10c indicate the direction of a potential rotation (red/blue) or whether no rotation is expected (gray).

**Circular Molecule.** Fig. 10a depicts the response of the circular molecule to manipulations on the surface. Manipulating the molecule pushes it in the direction opposite to the point of manipulation. The expected translation depends on the distance from the molecule's center: the largest movements occur when manipulated at a radius close to $0.9\,\text{nm}$, with shorter displacements when manipulated either closer or further away than $0.9\,\text{nm}$ from the center. The covariance $\sigma_{x,y}$ for each action $(x, y) \in \mathcal{A}$ for this molecule is set to $0.025I$. This induces random displacements from the mean translation $\mu_{x,y}$ of at most $L$ with $95\%$ probability. Due to its circular symmetry, the molecule does not rotate.

**Cross-shaped Molecules with $\sigma_{x,y} \in \{0.025I, 0.1I\}$.** The largest translations are expected when pulses are applied near the center of one of the molecule's arms, pushing it in the opposite direction. When manipulations are applied outside the cross-shaped outline, no movement is expected. For this type of molecule, we vary the isotropic covariance of the movement by setting it to $0.025I$ or $0.1I$ for all actions $(x, y) \in \mathcal{A}$. When choosing $\sigma_{x,y} = 0.025I$, random displacements from the mean translation $\mu_{x,y}$ by at most $L$ are expected with $95\%$. Conversely, when choosing $\sigma_{x,y} = 0.1I$, random displacements by at most $2.5L$ are expected with $95\%$. Due to this large covariance, the molecule's movement is dominated by random displacements, especially for manipulations with small mean translation $\mu_{x,y}$. These molecules can adopt discrete orientations $\rho \in \{0°, 30°, 60°\}$. Fig. 10c shows the corresponding rotational behavior. When a pulse is applied to

the left or right side of an arm, a clockwise or counterclockwise rotation occurs with probability $0.15$, indicated by blue and red dots, respectively. Manipulations close to the molecular center do not induce rotation.

### B.2. Assemblies

In the following, we describe the different types of assemblies used to evaluate our complete framework and show different examples of the assembly process in Figures 11-16. Each of the examples shows three images of the complete assembly process. The first image shows the initial distribution of molecules and obstacles on the surface, and the target configuration. This image also shows the initial, shortest matching, where assignments with precedence constraints are depicted by their corresponding corridors. Corridors drawn in red show constrained assignments that could not be scheduled. The second image shows the execution of the feasible schedule. The individual translations of each molecule are depicted using a color range from yellow for small movements to purple for movements that are close to $max_{move}$ of the molecule. The third image shows the constructed assembly.

**Honeycomb Assemblies.** The honeycomb assemblies are designed to evaluate our construction-plan algorithm on intricate assemblies. These assemblies are designed to investigate the interaction of molecules in assemblies that resemble graphene, which is of high interest for the surface science community.

For each honeycomb assembly, we start from a finite patch of a regular hexagonal lattice with $k$ rows and columns. At each node of this lattice, we place a regular hexagon, and take the anchors of these smaller hexagons as the target positions of the assembly. Across different instances, we vary only the parameter $k$, i.e., the number of rows and columns in the original hexagonal lattice. Fig. 9 in the main body shows a honeycomb assembly with $k = 1$, while Fig. 11, 12, and 13 depict assemblies with $k = 2$, $k = 3$, and $k = 5$, respectively. Because the target positions in these assemblies are densely packed on the surface, they provide a challenging benchmark for our construction-plan algorithm.

**Square Assemblies.** The square assemblies consist of target positions arranged along one large square. Fig. 14 and 15 show two examples, consisting of 36 and 180 molecules, respectively. The minimal distances between any two target positions in these assemblies is approximately 1.9nm. This allows us to test whether our approach is able to construct assemblies with molecules that are packed at close distances. Additionally, as the size of the surface grows with the size of the square, large assemblies test whether the trained RL policies are able to consistently keep molecules within their prescribed corridors along long paths.

**Quantum Corral.** As an additional assembly, we built the quantum corral as depicted in Figure 2 of (Correa et al., 2005). The target positions in this quantum corral are automatically computed, such that the resulting geometry realizes a desired interaction pattern between all molecules, providing an example of using automated spatial design to program quantum interactions.

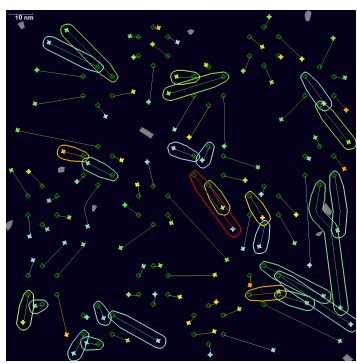 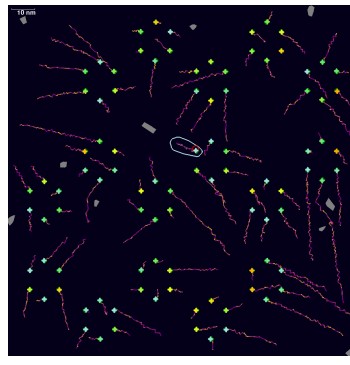 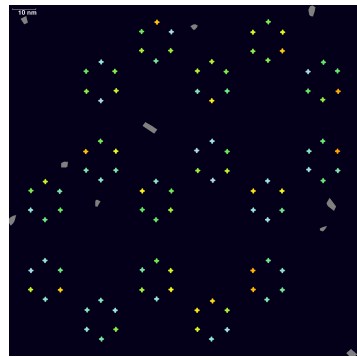

*(a)* The initial matching for this assembly, with assignments that could not be scheduled highlighted with a red corridor.

*(b)* Execution of the feasible schedule.

*(c)* The completed assembly.

*Figure 11.* The complete assembly process for a honeycomb assembly with $k = 2$ using the cross-shaped molecule with $\sigma_{x,y} = 0.025I$.

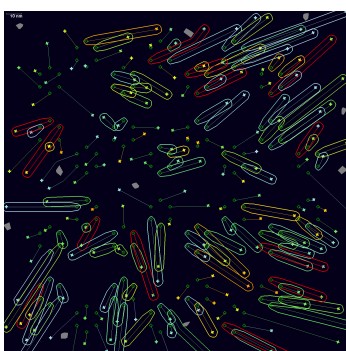 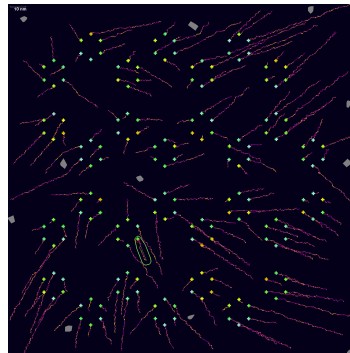 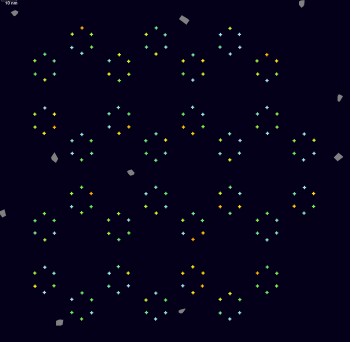

*(a)* The initial matching for this assembly, with assignments that could not be scheduled highlighted with a red corridor.

*(b)* Execution of the feasible schedule.

*(c)* The completed assembly.

*Figure 12.* The complete assembly process for a honeycomb assembly with $k = 3$ using the cross-shaped molecule with $\sigma_{x,y} = 0.025I$.

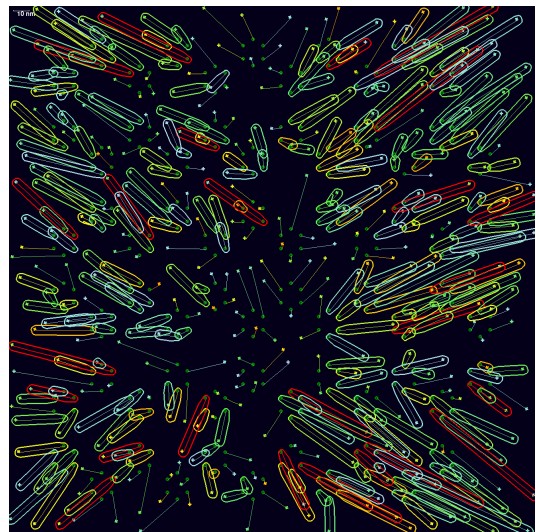

*(a)* The initial matching for this assembly, with assignments that could not be scheduled highlighted with a red corridor.

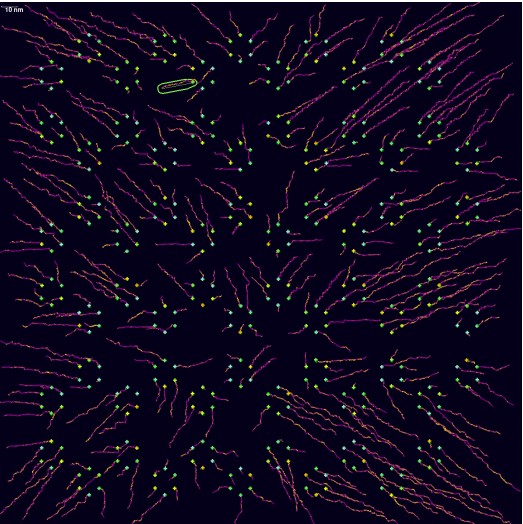

*(b)* Execution of the feasible schedule.

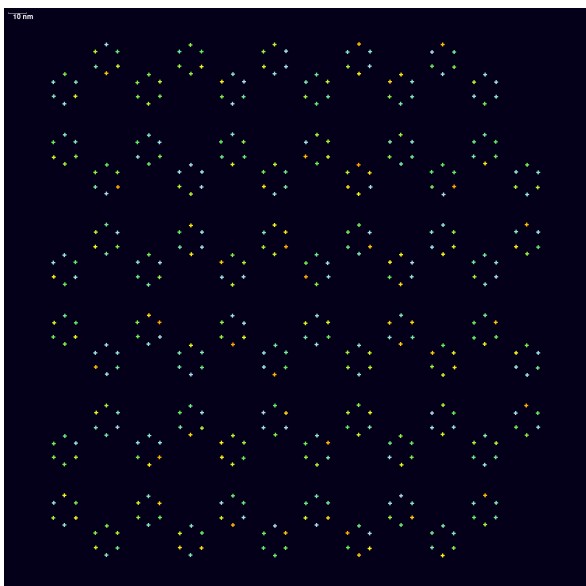

*(c)* The completed assembly.

*Figure 13.* The complete assembly process for a honeycomb assembly with $k = 5$ using the cross-shaped molecule with $\sigma_{x,y} = 0.025I$.

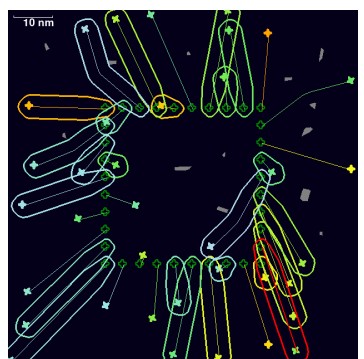

*(a)* The initial matching for this assembly, with assignments that could not be scheduled highlighted with a red corridor.

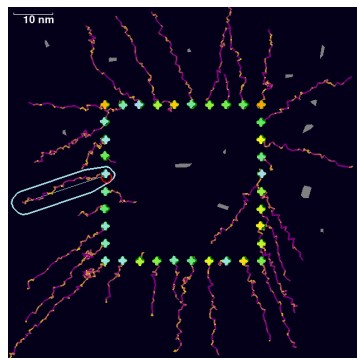

*(b)* Execution of the feasible schedule.

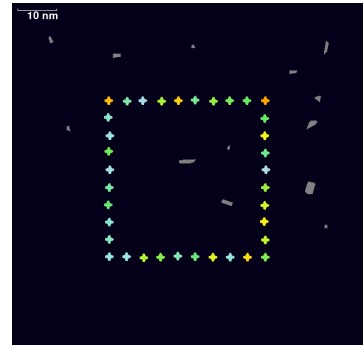

*(c)* The completed assembly.

*Figure 14.* The complete assembly process for a square assembly with 36 molecules using the cross-shaped molecule with $\sigma_{x,y} = 0.1I$.

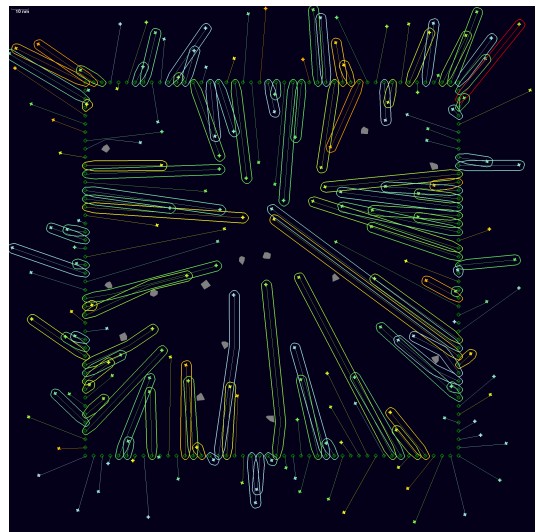

*(a)* The initial matching for this assembly, with assignments that could not be scheduled highlighted with a red corridor.

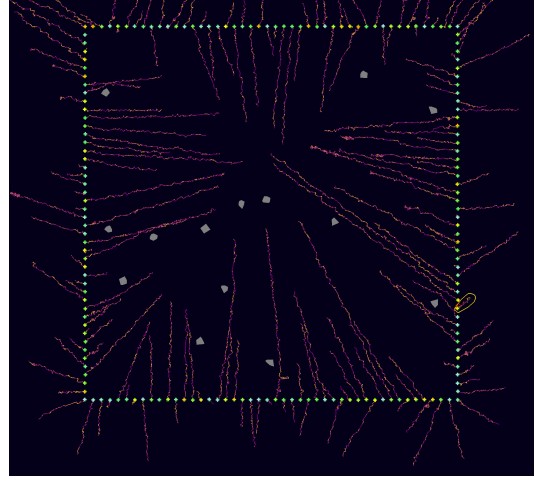

*(b)* Execution of the feasible schedule.

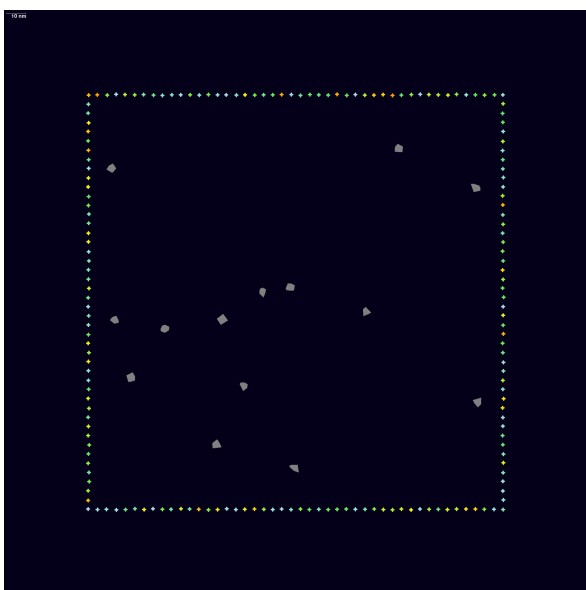

*(c)* The completed assembly.

*Figure 15.* The complete assembly process for a square assembly with 180 molecules using the cross-shaped molecule with $\sigma_{x,y} = 0.1I$.

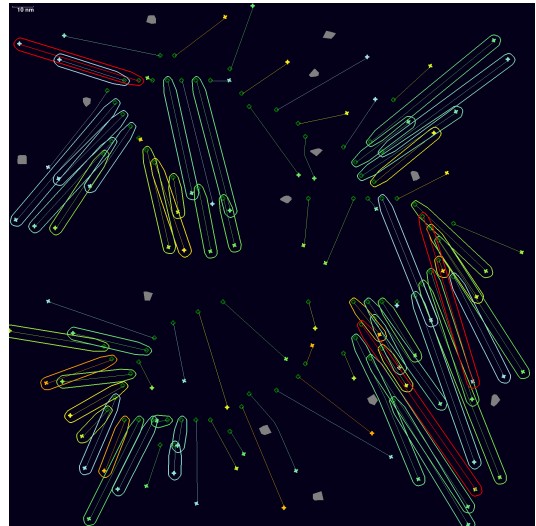

*(a)* The initial matching for this assembly, with assignments that could not be scheduled highlighted with a red corridor.

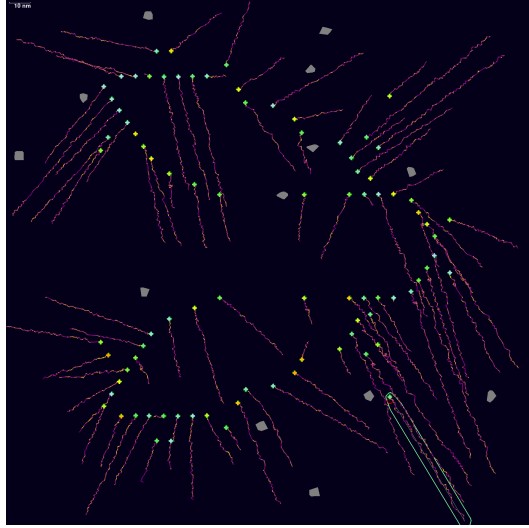

*(b)* Execution of the feasible schedule.

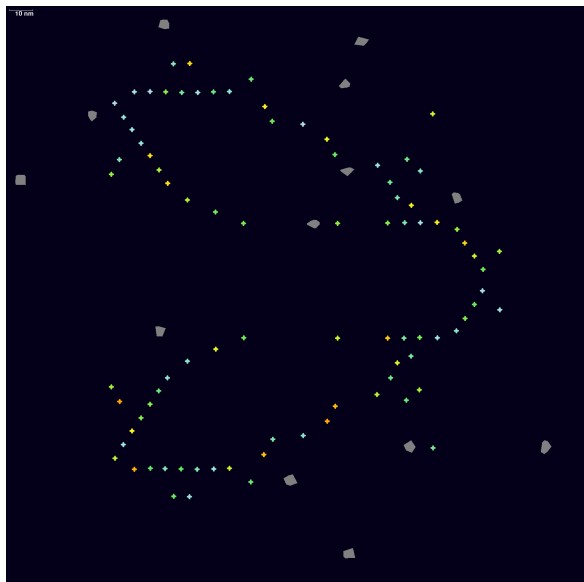

*(c)* The completed assembly.

*Figure 16.* The complete assembly process for the quantum corral using the cross-shaped molecule with $\sigma_{x,y} = 0.025I$.

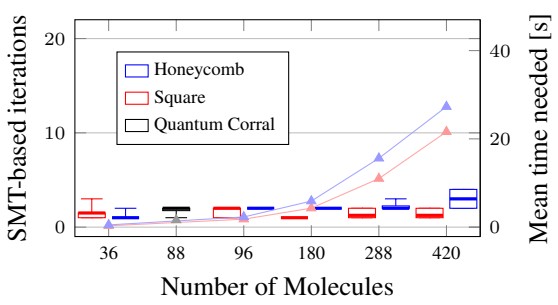
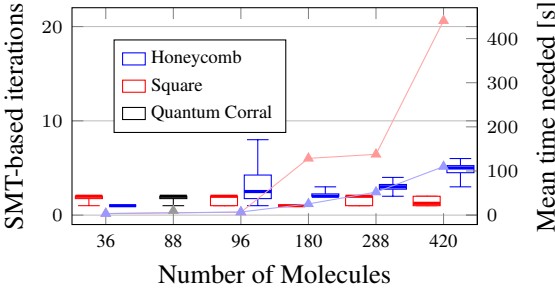

*(a)* Results for assembly construction without obstacles.

*(b)* Results for the assembly construction with obstacles.

*Figure 17.* The number of iterations (boxplots, left axis) and mean computation time for a feasible schedule (right axis) for the individual assemblies with $w = 4$ nm. Boxplots show the median (thick line), the $25^{th}$–$75^{th}$ percentiles (box), and the min. and max.

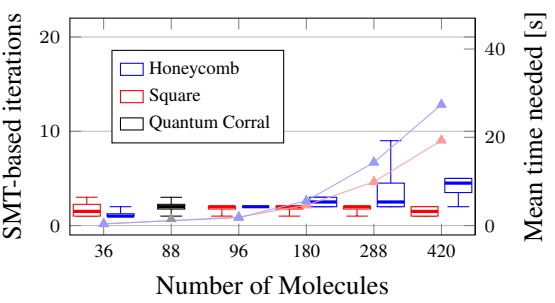
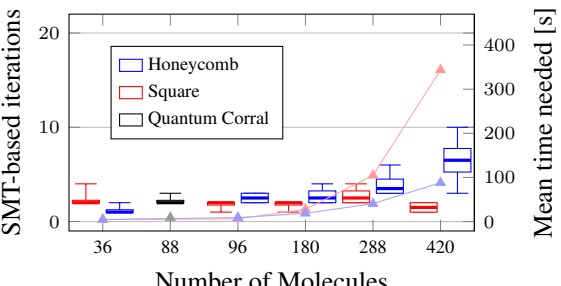

*(a)* Results for assembly construction without obstacles.

*(b)* Results for the assembly construction with obstacles.

*Figure 18.* The number of iterations (boxplots, left axis) and mean computation time for a feasible schedule (right axis) for the individual assemblies with $w = 5$ nm. Boxplots show the median (thick line), the $25^{th}$–$75^{th}$ percentiles (box), and the min. and max.

## C. Influence of Corridor Width on the Construction Plans

We computed construction plans for all assemblies, with and without obstacles, using corridor widths $w \in \{4, 5, 6\}$ nm. Figures 7, 17, and 18 show the corresponding results. A comparison across corridor widths shows a clear effect on the computation of construction plans: the narrower the corridor, the less iterations are needed to resolve conflicts. For a corridor width of $w = 4$ nm, construction plans can be successfully built within at most 5 iterations, with a single outlier requiring 9 iterations. Similar behavior is observed for $w = 5$ nm, where most plans for assemblies with less or equal than 180 molecules are computed in fewer than 5 iterations. Only the larger honeycomb assemblies require between 3 and 10 iterations.

The results for $w \in \{4, 5\}$ nm shown in Figures 17 and 18 confirm the effects of structural complexity and the time required to compute circumventing corridors. As in the case of $w = 6$ nm, square assemblies are typically resolved in few iterations, whereas honeycomb assemblies require a larger number of iterations due to their more complex structure. For $w \in \{4, 5\}$ nm, the results obtained in the presence of obstacles indicate that the corridor width has little influence on the time required to compute construction plans. This further supports the observation that the computation of circumventing corridors dominates the overall computation time.

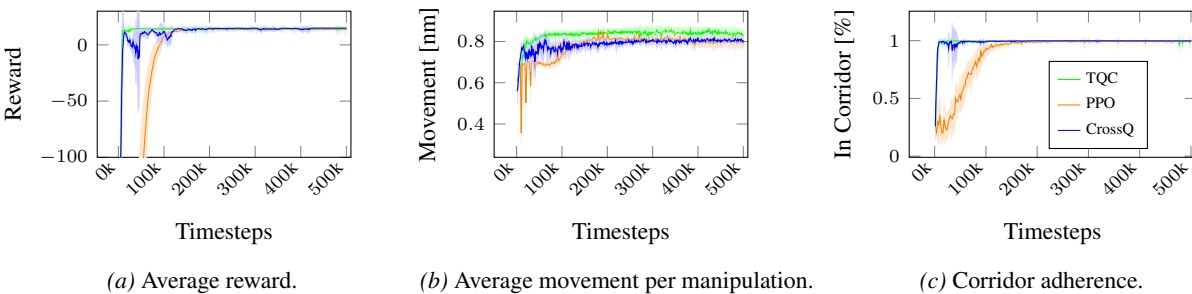

*(a) Average reward.*        *(b) Average movement per manipulation.*        *(c) Corridor adherence.*

*Figure 19.* Training results for the circular molecule with corridor width $w = 6$ nm averaged over 10 runs. The mean across runs is shown as a thick line, and the standard deviation as a shaded region.

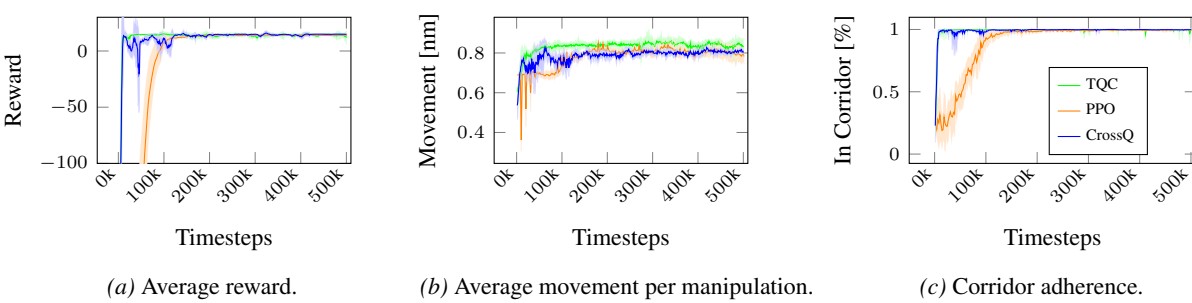

*(a) Average reward.*        *(b) Average movement per manipulation.*        *(c) Corridor adherence.*

*Figure 20.* Training results for the circular molecule with corridor width $w = 5$ nm averaged over 10 runs. The mean across runs is shown as a thick line, and the standard deviation as a shaded region.

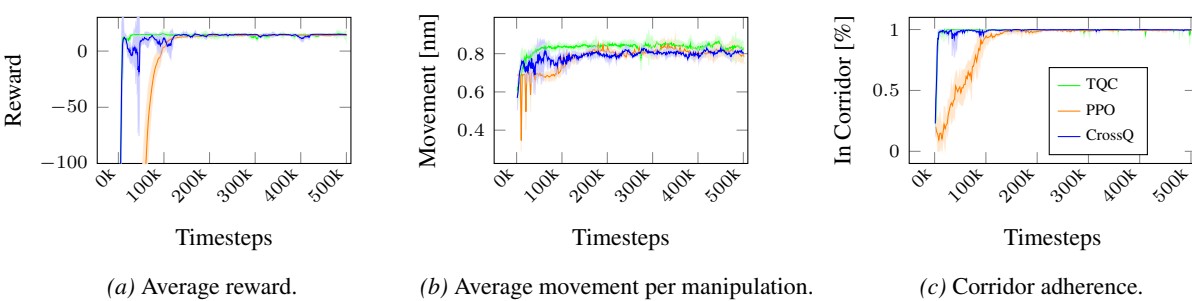

*(a) Average reward.*        *(b) Average movement per manipulation.*        *(c) Corridor adherence.*

*Figure 21.* Training results for the circular molecule with corridor width $w = 4$ nm averaged over 10 runs. The mean across runs is shown as a thick line, and the standard deviation as a shaded region.

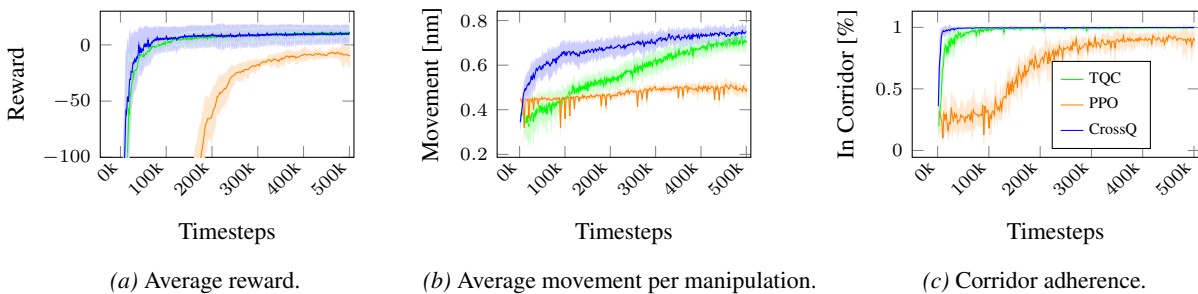

*(a)* Average reward.  *(b)* Average movement per manipulation.  *(c)* Corridor adherence.

*Figure 22.* Training results for the cross-shaped molecule with $\sigma_{x,y} = 0.025I$ with corridor width $w = 6$ nm averaged over 10 runs. The mean across runs is shown as a thick line, and the standard deviation as a shaded region.

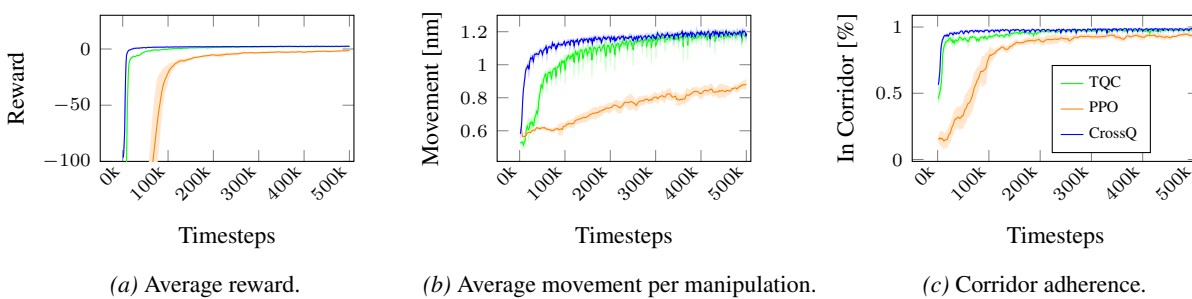

*(a)* Average reward.  *(b)* Average movement per manipulation.  *(c)* Corridor adherence.

*Figure 23.* Training results for the cross-shaped molecule with $\sigma_{x,y} = 0.025I$ with corridor width $w = 5$ nm averaged over 10 runs. The mean across runs is shown as a thick line, and the standard deviation as a shaded region.

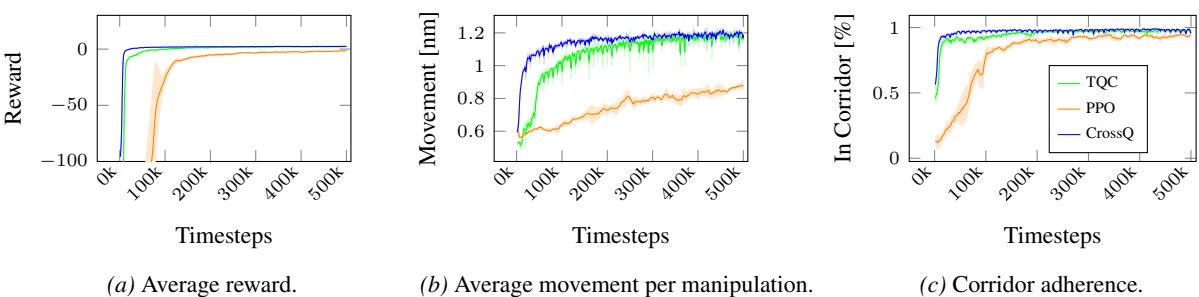

*(a)* Average reward.  *(b)* Average movement per manipulation.  *(c)* Corridor adherence.

*Figure 24.* Training results for the cross-shaped molecule with $\sigma_{x,y} = 0.025I$ with corridor width $w = 4$ nm averaged over 10 runs. The mean across runs is shown as a thick line, and the standard deviation as a shaded region.

# D. Details for RL Training

In the following, we discuss the RL training setup, the observation space used, and the training results for all types of molecules. We use a superscript index $t$ to indicate the current position $p_i^t$ and orientation $\alpha_i^t$ in timestep $t$.

**Training Setup.** We trained manipulation policies using CrossQ, TQC, and PPO implemented in Stable-Baselines3 (Raffin et al., 2021). For all three algorithms, we use a multi-layer perceptron policy and the default Stable-Baselines3 hyperparameters. The policies are trained to move the molecule efficiently in corridors with width $w \in \{4, 5, 6\}$ nm. Each policy is trained for a total of 500,000 timesteps. Episodes terminate either when the molecule reaches the target position and orientation or a maximum horizon of 100 steps is reached. The underlying Markov decision process $\mathcal{M}$ is described in Sec. 3.3, with its probabilistic transition function depending on the chosen molecule. The size of the action spaces is equal for all molecules, as described in Appendix B.1. For the reward function, we set $\varepsilon_{rot} = 1.0$ nm, such that orientation is only rewarded or penalized when the distance between the current position $p_i^t$ and the target position $q_j$ is below $1.0$, nm. The training environment is shown in Fig. 6b: the initial position of the molecule is fixed, while the initial orientation is sampled from all admissible orientations the molecule can adopt on the surface.

**Observation Space.** At each timestep $t$, the environment returns an observation vector $obs_t \in \mathbb{R}^7$ that encodes the current state of the molecule, its relation to the target state and its offset from the corridor center.

- $d(p_i^t, q_j)^{-1} \cdot \overrightarrow{p_i^t q_j}$, the normalized vector from $p_i^t$ to $q_j$,
- $\min(6, d(p_i, q_j))$, the current distance to the target, capped at 6nm,
- *lateral offset*, the signed lateral distance to the corridor centerline, clipped to $[-3, 3]$,
- $p_i^t \subset \texttt{Corr}(p, q)$, whether the molecule is within the corridor,
- $\alpha_i^t$, the molecule's orientation, and
- $\beta_j - \alpha_i^t$, the difference to the goal orientation.

We define the signed lateral offset as follows. Let $\ell = \overrightarrow{p_i q_j}$ denote the centerline of the corridor. Let $C$ be the orthogonal projection of the current position $p_i^t$ onto $\ell$, i.e.,

$$C = \arg\min_{x \in \ell} \|x - p_i^t\|_2.$$

Let $\mathbf{t} = \overrightarrow{t_x, t_y}$ be a (unit) tangent vector of $\ell$ at $C$, and let $\mathbf{v} = \overrightarrow{v_x, v_y} = p_i^t - C$ be the offset vector from the centerline to the molecule. The sign of the lateral offset is determined by the 2D cross product

$$\text{sign}(\mathbf{t}, \mathbf{v}) = \begin{cases} +1, & \text{if } t_x v_y - t_y v_x \geq 0, \\ -1, & \text{otherwise,} \end{cases}$$

and the clipped and signed lateral offset is

$$\text{clip}\big(\text{sign}(\mathbf{t}, \mathbf{v}) \, \|\mathbf{v}\|_2, , -3, 3\big),$$

with $\text{clip}(x, a, b) = \max\{a, \min\{b, x\}\}$.

If the molecule needs to circumvent an obstacle, it moves along the shortest path $\tau = p_i, v_1, v_2, \ldots, q_j$, between $p_i$ and $q_j$. Accordingly, it needs to move within corridors $\texttt{Corr}(p_i, v_1), \texttt{Corr}(v_1, v_2), \ldots, \texttt{Corr}(v_m, q_j)$. The observation $obs_t$ depends on the current corridor $\texttt{Corr}(v_k, v_{k+1})$ the molecule currently traverses. The direction the molecule needs to move to is changed to be $d(p_i^t, v_{k+1})^{-1} \cdot \overrightarrow{p_i^t v_{k+1}}$, i.e. pointing from the current position $p_i^t$ to the anchor $v_{k+1}$ of the corridor that is currently being traversed. To compute the lateral offset from the centerline of the currently traversed corridor, the definition of $\ell$ is changed to $\ell = \overrightarrow{p_i^t v_{k+1}}$ accordingly. When the distance $d(p_i^t, v_{k+1})$ between to molecule's position and $v_{k+1}$ is smaller than 1.0nm, $k$ is incremented to the next waypoint and the computation of the observation is changed accordingly.

**Training Results.** In the following, we discuss the training results for all types of molecules and justify the choice of policy used for evaluation. The training results for the circular molecule are shown in Figures 19–21, for the cross-shaped molecule with $\sigma_{x,y} = 0.025I$ in Figures 22–24, and for the cross-shaped molecule with $\sigma_{x,y} = 0.1I$ in Figures 25–27. All curves are averaged over 10 runs and reported as a moving average. The mean across runs is shown as a thick line, and the standard deviation as a shaded region. At the start of each training, all policies first need to learn to adhere to the corridor constraints and therefore receive large penalties. The plots that show the average reward are therefore limited to the range of $[-100, 15]$ to allow for comparison.

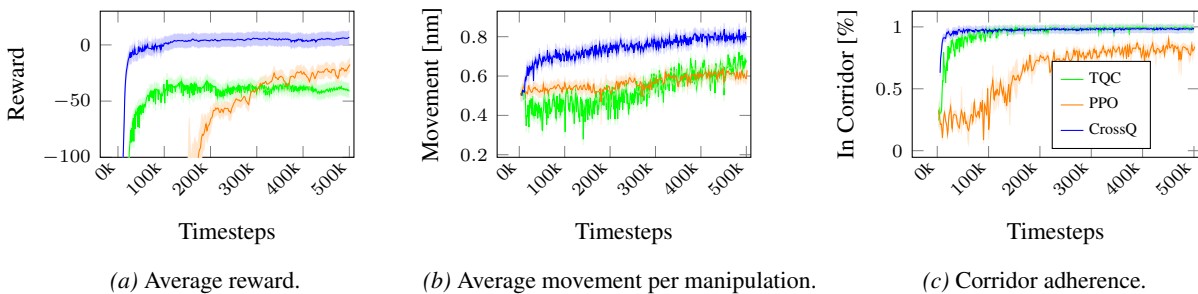

*(a) Average reward.*   *(b) Average movement per manipulation.*   *(c) Corridor adherence.*

*Figure 25.* Training results for the cross-shaped molecule with $\sigma_{x,y} = 0.1I$ with corridor width $w = 6\,\text{nm}$ averaged over 10 runs. The mean across runs is shown as a thick line, and the standard deviation as a shaded region.

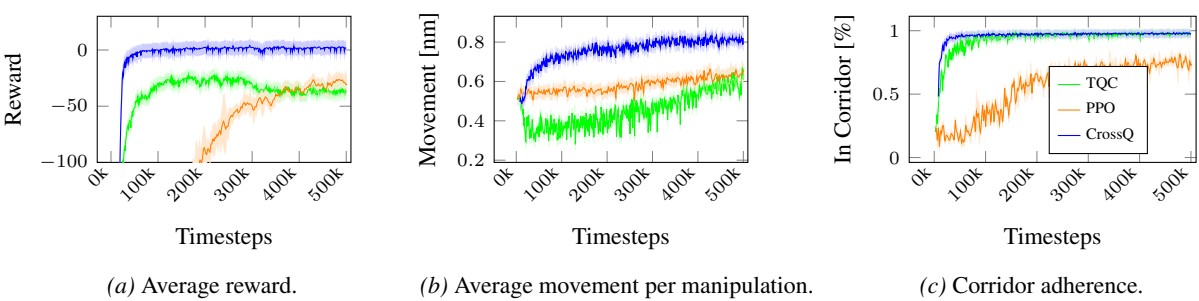

*(a) Average reward.*   *(b) Average movement per manipulation.*   *(c) Corridor adherence.*

*Figure 26.* Training results for the cross-shaped molecule with $\sigma_{x,y} = 0.1I$ with corridor width $w = 5\,\text{nm}$ averaged over 10 runs. The mean across runs is shown as a thick line, and the standard deviation as a shaded region.

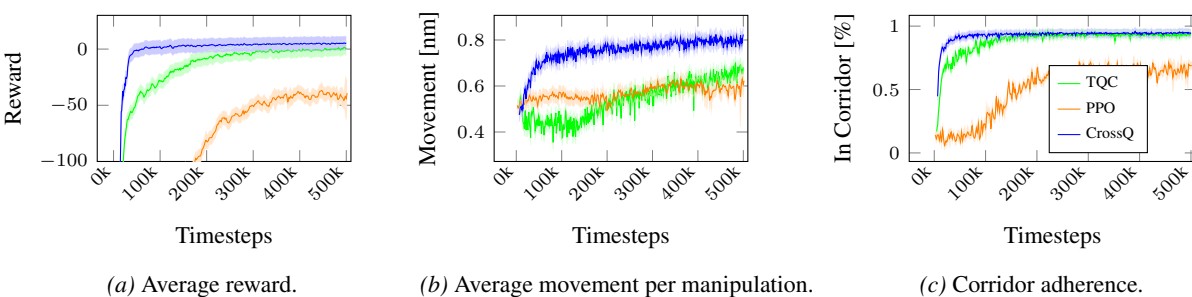

*(a) Average reward.*   *(b) Average movement per manipulation.*   *(c) Corridor adherence.*

*Figure 27.* Training results for the cross-shaped molecule with $\sigma_{x,y} = 0.1I$ with corridor width $w = 4\,\text{nm}$ averaged over 10 runs. The mean across runs is shown as a thick line, and the standard deviation as a shaded region.

**Circular Molecule.** For this molecule, the results show that all algorithms achieved favorable average rewards, independent of the corridor width $w$. Moreover, all algorithms managed to keep the molecule within the boundaries of the corridor during training. The average movement per manipulation indicates that policies trained with TQC are the most efficient, consequently, all assembly constructions were executed using a policy trained with TQC.

**Cross-shaped Molecule with $\sigma_{x,y} = 0.025I$.** For the cross-shaped molecule with $\sigma_{x,y} = 0.025I$, we observe that CrossQ outperformed both TQC and PPO. Policies trained using PPO fail to move the molecule within the corridor and do not manage to efficiently move the molecule either. CrossQ and TQC both manage to move the molecule within the corridor and achieve comparable rewards, where policies using CrossQ achieve slightly better rewards than policies trained with TQC. Policies that are trained with CrossQ manage to move the molecule more efficiently and all assembly constructions were therefore executed with a policy trained with CrossQ.

**Cross-shaped Molecule with $\sigma_{x,y} = 0.1I$.** For the cross-shaped molecule with $\sigma_{x,y} = 0.1I$, CrossQ clearly outperforms both TQC and PPO. As above, PPO fails to move the molecule within the corridor constraints for any corridor width $w \in \{4, 5, 6\}$ nm. After convergence, both CrossQ and TQC successfully move the molecule within the corridors in $95\%$ of all manipulations for a corridor width of $4$ nm, with success rates increasing to $98.3\%$ and $98.7\%$ for corridor widths of $5$ nm and $6$ nm, respectively. A comparison of efficiency after convergence shows that CrossQ consistently outperforms TQC across all corridor widths. For $w = 4$ nm, CrossQ achieves an average movement of $0.84$ nm compared to $0.68$ nm for TQC. For $w = 5$ nm, CrossQ achieves $0.81$ nm, while TQC reaches $0.60$ nm, and for $w = 6$ nm, CrossQ attains $0.80$ nm versus $0.66$ nm for TQC.

## E. Corridor Types

The algorithm to compute construction plans is agnostic to the concrete corridor shape and only requires each $\texttt{Corr}(p_i, q_j)$ to be a convex polygon, as defined in Sec. 3.2. In the following, we compare two possible ways to construct corridors. We write $\text{conv}(P)$ for the convex hull of a polygon $P$.

**Simple Corridor.** Based on the user-defined *corridor width* $w$, the corridor for assignment $a_{ij}$ can be computed as the Minkowski sum of $\overrightarrow{p_i q_j}$ and the Euclidean ball with radius $r = w/2$:

$$\texttt{Corr}(p_i, q_j) = \text{conv}(\overrightarrow{p_i q_j} \oplus \mathcal{B}_{w/2}(0, 0)).$$

This allows $m_i$ to be moved within a safe corridor and gives ample space to orient $m_i$ correctly at its target position, while implicitly limiting the minimal distance between any two points $q_j, q_k$ in the assembly $\texttt{Ass}$.

**Corridor with Narrow Parking.** To remedy the limitation of the first approach, we propose corridors that decrease in width close to the target position, as depicted in Fig. 5. Such corridors limit the space the molecule is allowed to use when *parking in* at its target position. In addition to the corridor width $w$, the corridors are computed based on the size of a *buffer* $b$ around the molecule at its target position and the distance $dist$ from which the corridor needs to decrease in width in order to meet the buffer $b$. Such corridors are computed as

$$\begin{aligned} \texttt{Corr}(p_i, q_j) = \text{conv} \big[ &(d(p_i, q_j) - dist) \cdot \overrightarrow{p_i q_j} \oplus \mathcal{B}_{w/2}(0, 0) \\ &\cup\, q_j \oplus \mathcal{B}_b(0, 0) \big] \end{aligned}$$

Depending on the expected movement when manipulating a given type of molecule, modifying $b$ allows to compute strategies with small distances between goal positions in $\texttt{Ass}$.

