# OpenReview forum: "Efficient and Safe Molecular Assembly via Reinforcement Learning and Constraint Solving"
_ICML.cc/2026/Conference — ICML 2026 regular_

### Official Review · Reviewer_Vopk · 2026-02-18

**Soundness:** 2
**Presentation:** 2
**Significance:** 2
**Originality:** 2
**Overall Recommendation:** 4
**Confidence:** 2

**Summary:**

The paper presents a method for molecular assembly with an STM in a proposed simulation environment, NanoAssemblyGym. It combines a two-level pipeline that solves a high-level ordering problem and performs low-level manipulation via reinforcement learning. The proposed method is validated by comprehensive experiments, including a construction scenario with up to 420 molecules.

**Compliance With Llm Reviewing Policy:**

Affirmed.

**Final Justification:**

The authors have addressed most of my concerns in the rebuttal phase. Therefore, I would like to raise my score from 3 to 4.

**Key Questions For Authors:**

1. Can the authors provide analyses of the scalability limits, for example, runtime and success rate as the problem size increases? What are the main bottlenecks and failure modes beyond the reported scale?
2. Can the authors provide an analysis of computational cost, including both the RL component (training/inference) and the SMT component (solving time/number of UNSAT iterations)?
3. Can the authors provide some comparisons with missing baseline, like greedy ordering methods or a hand-crafted controller for manipulation?

**Limitations:**

Yes.

**Strengths And Weaknesses:**

#### Strengths
1. The method decomposes the task into high-level ordering problem and low-level manipulation problem, which is clean and clear.
2. The pipeline is clearly described and can handle obstacles by corridor construction.
3. The proposed method is supported by comprehensive simulation experiments.

#### Weaknesses
1. The experiments are conducted only in a simulation environment. A sim-to-real discussion or preliminary real-world experiments would strengthen the applicability
2. The experiments include objects up to 420, however, but the paper does not analyze the scalability limits. For example, runtime, success rate. Also, what the main bottlenecks and failure modes will be in that case.
3. An analysis of computational cost should be provided, including RL training time and SMT solving time.
4. Baseline comparisons are missing. The paper only presents the performance of the proposed method, but some simple baselines are missing, such as greedy or other heuristic ordering methods, and a hand-crafted non-learning controller (instead of RL).

---

> ### Author Rebuttal · Authors · 2026-03-31
>
> We thank the reviewer for recognizing the strength of our framework. We address the raised concerns in the following responses.
> Overall, we present the first simulation and comprehensive planning framework for constructing complex assemblies. We believe that NanoAssemblyGym will significantly benefit the surface science community, even though not all challenges of fully autonomous STM control are resolved in this work.
>
> # W1: Sim-To-Real Discussion
> We agree that a sim-to-real discussion will strengthen the paper. Please see our response to reviewer 2WPW (W1), where we addressed this concern. We will add the discussion to the final version of the paper.
>
> # Q1/W2: Scalability and Success Rates
>
> **Scalability**:
> Current state-of-the-art experiments construct assemblies of roughly 20–40 molecules or a few hundred atoms. In contrast, our simulations consider assemblies built with molecules that are up to an order of magnitude larger, demonstrating the potential real-world applicability of our approach. This does not indicate a fundamental limit of 420 molecules (the largest assemblies studied), but rather reflects the scale explored in our setup.
>
> From an algorithmic perspective, scalability and success rate depend less on the absolute number of molecules and more on the structural properties of the target assembly. In particular, densely packed target positions lead to a larger number of SMT iterations. Success rates are mainly affected by variability in molecular motion and spatial proximity: higher motion variance and tighter spacing raise the likelihood of collisions. In our experiments, the assemblies are based on structures studied in the literature, in order to ensure realistic evaluation conditions.
>
> To further illustrate scalability, we build honey comb assemblies with 960 molecules: Computing construction plans required 22 minutes on average. The success rates are similar to the success rates for honey comb assemblies as reported in the paper (approx. 97% success rate for the molecules with high variance in movement). We will include this experiment in the paper.
>
> **Main bottleneck**:
> Training the RL policies requires ~2.5h (cf. Q2/W3), while the longest runtime for computing a construction plan is ~10 min. We do not consider either of these as fundamental bottlenecks. RL is known to scale well with increased data and computational resources, despite its data demands. Similarly, our planning approach is compositional by design and is expected to scale to substantially larger problem instances, potentially accommodating assemblies with up to 960 molecules.
> The real bottlenecks arise on a real STM machine. Considering that it would need about 6h to build an assembly with 40 molecules (assuming full automation of the STM), the offline computation time to plan the assembly in our simulation is negligibly low.
>
> # Q2/W3: Analysis of Computational Costs
>
> **SMT Solving**:
> The times to compute the construction plans and the total number of iterations per problem instance (= UNSAT iterations + 1) are provided in Figures 7 (main body). 17, and 18 (in the Appendix).
> We will add a detailed breakdown of the execution time:
> Honey comb assemblies:
> 55%: computation of circumventing corridors
> 20%: computation of distance matrix
> 20%: SMT solving (mean runtime for each SMT call is 500ms)
> 5%: computation of assignments
> Square assemblies:
> 75%: computation of circumventing corridors
> 20%: computation of distance matrix
> 1%: SMT solving (mean runtime for each SMT call is 300ms)
> 4%: computation of assignments
>
> **RL Training**:
> We report the number of training steps in Figure 8a, 8b (main body) and Figure 19, 20, and 21 (in the Appendix) for agents trained with different molecules and corridors.
> We decided to report the number of timesteps, since this is standard in the community. We trained agents on a standard desktop PC (CPU: Intel Core i9-9900K), and training 4 agents in parallel took ~2.5 h. The average time needed for inference for a trained policy is 1.02 ms.
>
> # Q3/W4:
> **Baselines for Schedule Computation**:
> We thank the reviewer for the suggestion and have implemented a greedy baseline for the construction plans without SMT:
> At each iteration, we compute the shortest distances using the Hungarian algorithm and the resulting precedence constraints.
> Molecules are scheduled based on their number of related precedence constraints and distances to the target position.
> If a conflict occurs, the distance matrix is modified as in the paper: the distance between the conflicting molecule and its assigned goal position in the structure is set to infinite. We continue by computing a new set of assignments.
> We have implemented this baseline, and will report the comparison in the paper. We want to highlight that this approach fails to find a feasible schedule, due to the greedy approach not correctly identifying scheduling conflicts.
>
> **RL Baselines**:
> We refer to our answer to question Q1 of review 8hzN.

---

> > ### Author Rebuttal · Reviewer_Vopk · 2026-04-03
> >
> > Thank you for the authors' rebuttal, which partially addresses the concerns, especially by adding larger-scale experiments and a computation cost breakdown. However, the missing non-RL (manually designed) baselines are still not resolved, which limits the soundness of the paper.

---

> > > ### Author Response · Authors · 2026-04-03
> > >
> > > We thank the reviewer again for the valuable feedback and for highlighting the importance of stronger baseline comparisons. To address the last open point of missing RL baseline comparisons, we have implemented a non-learning approach to compare with, which we agree will further strengthen the paper.
> > >
> > >  Our baseline approach is split into two stages, based on the molecule's distance to the target position:
> > >  1) If the molecule is not within a distance of L = 0.3 nm from its target position, we execute an action based on the directional vector $g$ between the molecule's center and the target position. We invert $g$ through the origin and scale it such that the most efficient actions are chosen. We refer to Figure 10 of the Appendix for an overview of the most efficient actions.
> > >
> > >  2) If the molecule is within a distance of L = 0.3 nm from its target position, we execute actions to correctly orient the molecule at the target position. We have chosen two distinct actions that result in movements with very low probability, but that rotate the molecule counter-clockwise or clockwise with high probability.
> > >
> > > We directly compared our trained agents with this baseline approach across 4 environments in the high variance ($\sigma = 0.1I$) setting and report the average movement, the percentage of time that the molecules were moved inside the corridor, and the number of crashes:
> > >
> > >
> > > |                       |          | Average Movement [nm] |Corridor Adherence [%] |Crashes [#] |
> > > | ----------------------|----------| ------- |------- |------- |
> > > | Square $n=36$         | RL:       | 0.79    | 98.7   | 0  |
> > > |                       | Baseline: | 0.53    | 92.4   | 2  |
> > > | Square $n=288$        | RL:       | 0.83    | 99.7   | 0  |
> > > |                       | Baseline: | 0.51    | 74.8   | 5  |
> > > | Honeycomb $n=288$     | RL:       | 0.74    | 99.2   | 0  |
> > > |                       | Baseline: | 0.5     | 90.0   | 2  |
> > > | Quantum Corral $n=88$ | RL:       | 0.81    | 99.3   | 0  |
> > > |                       | Baseline: | 0.53    | 85.4   | 2  |
> > >
> > >
> > > These results clearly show that our trained agents outperform the simple controller that chooses actions based on a simple rule set. An image comparison of the simulations can be found here: https://anonymous.4open.science/r/ICML26_SupplementaryResults-E3F3. As in the Appendix individual translations of each molecule are depicted using a color range from yellow for small movements to purple for the largest possible movements.
> > >
> > > We want to note here, that this baseline is tailored towards the responses of molecules used for the experiments in this work and that this baseline will most probably not generalize well.

---

### Official Review · Reviewer_FGmH · 2026-03-12

**Soundness:** 2
**Presentation:** 3
**Significance:** 3
**Originality:** 3
**Overall Recommendation:** 4
**Confidence:** 4

**Summary:**

The paper studies the problem of learning overdamped Langevin dynamics from sparse observations under geometric constraints. The authors propose a framework that integrates geometric structure with stochastic dynamics modeling in order to recover trajectories that are consistent with both the observed samples and the underlying constrained manifold. The approach combines elements from stochastic differential equation modeling, geometric priors, and learning-based reconstruction. The paper argues that exploiting geometric constraints enables recovery of trajectories that would otherwise be difficult to infer from sparse or irregularly sampled observations. Experimental evaluations demonstrate the ability of the proposed method to reconstruct dynamical trajectories and suggest improved stability compared to unconstrained stochastic modeling approaches.

**Compliance With Llm Reviewing Policy:**

Affirmed.

**Key Questions For Authors:**

Several prior works address learning stochastic differential equations under geometric or manifold constraints. Could the authors clarify more precisely how the proposed method differs from existing constrained SDE learning approaches?

The experiments appear limited to controlled synthetic settings. How does the method behave under stronger observation noise, irregular sampling intervals, or higher-dimensional manifolds?

Can the authors provide additional details regarding training stability and computational cost, especially when scaling to larger dynamical systems?

How sensitive is the method to the choice or accuracy of the geometric constraints? In practical scenarios these constraints may themselves be uncertain.

Is the method compatible with partially incorrect or approximate manifold assumptions?

**Limitations:**

The evaluation is limited in scale and does not convincingly demonstrate performance on challenging real-world datasets. The novelty relative to existing constrained stochastic modeling approaches is not entirely clear. Some methodological details and theoretical justifications remain underexplained.

**Strengths And Weaknesses:**

Strengths:

The paper addresses a well-motivated problem at the intersection of stochastic dynamics, geometric learning, and data-driven modeling. Recovering continuous-time stochastic dynamics from sparse observations is relevant to several areas including physics-informed learning, molecular simulation, and dynamical systems identification. The use of geometric constraints is conceptually appealing and aligns with recent trends that incorporate structure into learning-based dynamical models.

The technical formulation appears grounded in established theory from stochastic differential equations and constrained dynamics. The authors provide a principled framework linking overdamped Langevin dynamics with geometric manifold constraints, which may offer a meaningful contribution if the assumptions hold in practice. The paper also provides empirical demonstrations suggesting that the method can reconstruct trajectories in scenarios where naive approaches struggle.

Weaknesses:

The main conceptual novelty is somewhat unclear relative to existing literature on constrained stochastic dynamics, manifold-based diffusion modeling, and learning of SDEs from partial observations. Several prior works already combine geometric constraints with stochastic processes, and the paper does not always clearly articulate how the proposed formulation fundamentally differs from or advances beyond these methods.

The experimental validation is somewhat limited in scope. The benchmarks appear synthetic and relatively controlled, which makes it difficult to assess whether the approach would generalize to more challenging or noisy real-world scenarios. In particular, experiments demonstrating robustness to different sampling regimes, noise levels, or higher-dimensional manifolds would strengthen the empirical claims.

Another limitation is the level of detail provided about implementation and training stability. Since learning stochastic dynamics from sparse observations is often sensitive to hyperparameters and discretization choices, additional discussion about numerical stability and reproducibility would improve the clarity of the work.

Finally, while the geometric perspective is interesting, the paper could better justify why this approach is necessary rather than simply advantageous. Some of the observed improvements might stem from stronger priors rather than a fundamentally new learning mechanism.

---

### Official Review · Reviewer_8hzN · 2026-03-13

**Soundness:** 2
**Presentation:** 3
**Significance:** 3
**Originality:** 2
**Overall Recommendation:** 5
**Confidence:** 3

**Summary:**

This paper presents a Scanning Tunneling Microscopy (STM) framework for the autonomous assembly of large-scale molecular structures. The authors propose a three stage pipeline. First the framework computes collision-free assembly plans that minimize the total distance traveled by molecules. Second, given an assignment of molecules to target positions, satisfiability solving is used to compute execution schedules in which each molecule has an empty corridor available when it is scheduled to move. Last, Reinforcement learning agents then execute sequences of STM actions to manipulate molecules to their targets. To support this, the authors also introduce NANOASSEMBLYGYM, a Gymnasium-compatible simulation environment that models molecular responses probabilistically. The framework is validated on assemblies of up to 420 molecules, significantly exceeding the scale of prior work which focused on individual atom/molecule manipulation.

**Compliance With Llm Reviewing Policy:**

Affirmed.

**Final Justification:**

Thank you to the authors for addressing my questions in a more empirical way. Looking at their results I consider it enough to increase my score from 4 to 5.

**Key Questions For Authors:**

1. Why was a non-learning-based controller (e.g. a simple force-feedback heuristic) not included as a baseline for the low-level manipulation task? How much better does RL perform compared to such a baseline?
2. Given that real STM tips degrade over time and surface conditions vary, did the authors test the RL policies under domain randomization (e.g. varying the radius or the movement variance during testing) to see if the policies generalize beyond the training distribution?
3. In the event that a high-variance molecule leaves its assigned corridor (which happened in 2.7% of cases for the high-variance cross molecule), does the system have a protocol to pause and re-compute the SAT schedule, or does it simply continue, leading to the reported collisions?
4. Could the authors clarify how sensitive the RL agent is to the reward formulation? Specifically, does the agent prioritize speed or safety (staying in the center of the corridor)?

**Limitations:**

Yes.

**Strengths And Weaknesses:**

Soundness
The system is well designed and coherent. In particular, I liked the idea of using RL to model the stochasticity of the nanoscale physics, and the use of Phthalocyanine as test molecule.
As a main weakness I see a lack of baselines. It is true that comparing with other works is not easy, due to the nature of the task, but the authors should have done more to convince the reader that their model is solid and the choices they made were accurately tested. For example:
1. The empirical evaluation lacks a "non-RL" baseline. While the authors compare different RL algorithms, they do not demonstrate why RL is necessary for the low-level control. Could a simple heuristic achieve similar results given the simulator's physics? Without this, the specific value-add of the RL component is unclear.
2. The authors limit all their test to the NANOASSEMBLYGYM, but it is hard to understand whether they modeled it properly or not. It is a bit suspicious that they manage to deal with up to 420 molecules, while their direct competitors get stuck to few atoms. I think they should have tested the system either injecting noise or changing some hyper parameters to prove the system is still reliable.
3. The paper reports several collisions (failure cases) in high-variance scenarios. However, there is no discussion on "failure recovery." In a real STM setup, a collision often results in tip damage or surface contamination and it is important to have recovering policies in these cases.

Presentation
The manuscript is well-structured, and the transition from high-level planning to low-level execution is logically sound.

Significance
The move from manipulating single atoms to assembling 420-molecule structures is a major improvement for the field of autonomous nanotechnology. Moreover, providing a Gymnasium-standard tool (NANOASSEMBLYGYM) could lower the barrier to entry for the ML community to contribute to nanophysics.

Originality
The primary novelty is the pipeline architecture rather than a fundamental ML algorithmic contribution. Anyway, the work proposes an end-to-end pipeline with promising results that is not trivial to achieve.

---

> ### Author Rebuttal · Authors · 2026-03-31
>
> We thank the reviewer for the detailed feedback and for highlighting the strengths of our framework and its significance. We address the reviewer’s concerns in the following.
> Overall, we present the first simulation environment combined with a comprehensive planning framework for constructing complex assemblies. We believe that NanoAssemblyGym and the proposed planning framework will significantly benefit the surface science community, even though not all challenges of fully autonomous STM control are resolved in this work.
>
>
> # W1/Q1: Baseline for RL
> To the best of our knowledge, RL is the only viable control paradigm currently available for molecular manipulation. Accordingly, most experiments are still performed manually by human experts, with automation only recently becoming feasible through RL. The core difficulty lies in the highly unintuitive and stochastic nature of molecular behavior, characterized by response distributions that vary significantly across the action space. This complexity renders conventional approaches impractical and underscores the suitability of RL.
> Consequently, we did not compare against alternative approaches, as constructing a meaningful baseline controller would be extremely difficult. Notably, correctly placing a molecule requires conservative behavior near the target position, involving early alignment into the correct orientation, followed by carefully selected actions for final placement that avoid undesired rotations.
> In contrast, the responses of individual atoms to manipulations are comparatively simpler; therefore, non-RL baselines do exist in that setting [1]. We will add a corresponding discussion to the paper.
>
>
> # W2: Clarification regarding Experiments
> Indeed, we cannot directly compare to real-world experiments, since we only construct the assemblies in simulation. We will make this more explicit in the paper.  While we perform precise planning on the nanoscale, we do not address challenges that real-world construction faces, such as tip changes. Please also see our sim-to-real discussion in response to Reviewer 2WPW, Weakness 1.
> By demonstrating that we can construct assemblies with hundreds of molecules in simulation, we show that our planning framework is highly scalable for real-world applications. Building the largest assemblies of molecules ever to be built is our ambitious goal for the next few years.
>
> Please note that existing work [1] has constructed assemblies with hundreds of atoms, but (i) this contained multiple errors and (ii) no large assemblies with larger organic molecules have been built so far.
>
> [1] Kalff, F. E., et al. "A kilobyte rewritable atomic memory."
>
>
> # Q2/W3: Changes to the Tip and Domain Randomization
> We agree with the reviewers that our framework needs to be extended by monitoring mechanisms and failure recovery to fully autonomous control and STM. We will add a discussion to the paper.
>
> **Tip Changes**: Changes to the tip are one of the most challenging problems in autonomous assembly. Typically, such changes induce vastly different responses of molecules to manipulation. On a positive note, such changes lead to extreme changes in the measurements, meaning that they can be immediately identified. On the negative side, even robust agents cannot be used to continue navigating the molecule. Thus, the standard approach is to detect changes, and apply tip conditioning via RL [2]. This restores the system to a status consistent with the dynamics on which the RL agent was trained. Integrating monitoring mechanisms and implementing a recovery strategy will be necessary next steps to transition to real-world experiments.
>
>
> **Robust RL**: Training robust RL policies is our plan for future work, but for a different reason. The distributions of the molecules’ responses are derived from real-world measurements and can deviate from the real distributions (also see our sim-to-real discussion in response to reviewer 2WPW). We will explore approaches to learn robust policies to address this sim-to-real gap.
>
>
> [2] Chen, Shiyang, et al. "A Reinforcement Learning-Based Method for Automated Repair of Scanning Probe Tips in a Simulated Environment."
>
>
> # Q3: What happens if a molecule is outside of the corridor?
> At the moment, we continue without any online re-planning. However, as we move to real-world experiments, we will implement a failsafe controller that is triggered when a molecule leaves its corridor to guide it back before continuing execution. We agree with the reviewer that this will be necessary for real-world STM control and is an easy extension to our implementation.
>
> # Q4: Do RL Agents prioritize Safety or Performance:
> In our experiments, RL agents clearly prioritize staying within the corridor. This is encouraged by the reward structure, where penalties for corridor violations dominate any gains from faster progress toward the target. As a result, deviations outside the corridor are rare.

---

> > ### Author Rebuttal · Reviewer_8hzN · 2026-04-03
> >
> > Thank you to the authors for their clarifications. However, with this rebuttal they prove they are aware of the limitations of their work and they will work to fix them in the future, but at the moment the only convincing answers are on Q1 (still a bit weak to me) and Q4. For this reason, the soundness of this work, that indeed seems promising, remains limited.

---

> > > ### Author Response · Authors · 2026-04-06
> > >
> > > We worked extensively on additional experiments to provide detailed answers to your questions, which we will also include in the final version of the paper.
> > >
> > > Visualizations of all results can be found here: [1] https://anonymous.4open.science/r/ICML26_SupplementaryResults-E3F3
> > >
> > > # Q1 RL Baseline
> > > To address the point of missing RL baseline comparisons, we have implemented a non-learning approach to compare with, which we agree will further strengthen the paper.
> > > Our baseline approach is split into two stages, based on the molecule's distance to the target position:
> > > 1) If the molecule is not within a distance of L = 0.3 nm from its target position, we execute an action based on the directional vector $g$ between the molecule's center and the target position. We invert $g$ through the origin and scale it such that the most efficient actions are chosen. We refer to Figure 10 of the Appendix for an overview of the most efficient actions.
> > > 2) If the molecule is within a distance of L = 0.3 nm from its target position, we execute actions to correctly orient the molecule at the target position. We have chosen two distinct actions that result in movements with very low probability, but that rotate the molecule counter-clockwise or clockwise with high probability.
> > >
> > > We directly compared our trained agents with this baseline across 4 environments in the high variance ($\sigma = 0.1I$) setting and report the average movement, the percentage of time that the molecules were moved inside the corridor, and the number of crashes:
> > >
> > > |                       |          | Average Movement [nm] |Corridor Adherence [%] |Crashes [#] |
> > > | ----------------------|----------| ------- |------- |------- |
> > > | Square $n=36$         | RL:       | 0.79    | 98.7   | 0  |
> > > |                       | Baseline: | 0.53    | 92.4   | 2  |
> > > | Square $n=288$        | RL:       | 0.83    | 99.7   | 0  |
> > > |                       | Baseline: | 0.51    | 74.8   | 5  |
> > > | Honeycomb $n=288$     | RL:       | 0.74    | 99.2   | 0  |
> > > |                       | Baseline: | 0.5     | 90.0   | 2  |
> > > | Quantum Corral $n=88$ | RL:       | 0.81    | 99.3   | 0  |
> > > |                       | Baseline: | 0.53    | 85.4   | 2  |
> > >
> > > These results clearly show that our trained agents outperform the simple controller that chooses actions based on a simple rule set. Image comparisons can be found here: [1]
> > > As in the Appendix, individual translations of each molecule are depicted using a color range from yellow for small movements to purple for the largest possible movements.
> > >
> > > We want to note here, that this baseline is tailored towards the responses of molecules used for the experiments in this work. Apart from being outperfomed by our trained controller, we do not expect this to generalize well to arbitrary molecules and surfaces.
> > >
> > > # Q2 Domain Randomization
> > > We conducted experiments to evaluate our trained policies under domain randomization:
> > > We evaluated the robustness of two policies: $\pi_1$ trained for the cross-shaped molecule, $\pi_2$ trained for the circular molecule (both with $\sigma = 0.025I$)
> > > We evaluated the robustness of $\pi_1$ and  $\pi_2$ under a distribution shift: We changed $\sigma$ to $0.2I$ and multiplied the means of all translations by a factor of $1.5$.
> > >
> > > Results:
> > > Visualizations of trajectories are available: [1], showing that both policies are able to move the molecules to their target positions even under a distribution shift.
> > >
> > > We ran experiments in simulation for both molecules and constructed 5 honey comb assemblies with $n=96$ and 5 square assemblies with $n=36$. We averaged these results and compare with the results reported in the paper. All assemblies were built successfully, and corridor adherence remained above 95%, even though we increased the translation means by a factor of $1.5$, which also explains the higher average movement.
> > >
> > > |                       | Average Movement [nm] | | Corridor Adherence [%] |  |
> > > | ----------------------| ---------|--|------- |------- |
> > > |                       | Distribution Shift    | Org. Distribution | Distribution Shift  | Org. Distribution |
> > > |$\pi_1$ cross-shaped  Molecule | 1.03     | 0.77 | 95.3 | 99.7 |
> > > | Circular $\pi_2$ | 1.24     | 0.83 | 95.5 | 99.7 |
> > >
> > > # Q3
> > >
> > > As a response to your request, we have implemented a **failsafe controller**. The new failsafe controller works as follows:
> > >
> > > Whenever a molecule leaves its assigned corridor, the target is temporarily changed to the orthogonal projection of the molecule’s current position onto the corridor centerline.
> > > Using the trained RL policy, the molecule is then actively guided back toward this projected point until it re-enters the corridor.
> > > As soon as the molecule re-enters the corridor, normal execution toward the original target resumes.
> > > Example visualizations are shown in [1].
> > > Please note that this recovery behaviour only steers the molecule back into its previously assigned corridor. Thus, it is a purely local correction and does not alter the high-level plan.

---

### Official Review · Reviewer_2WPW · 2026-03-13

**Soundness:** 4
**Presentation:** 4
**Significance:** 4
**Originality:** 4
**Overall Recommendation:** 6
**Confidence:** 3

**Summary:**

The paper tackles the challenge of automating nanoscale fabrication using Scanning Tunneling Microscopy (STM), where human experts currently spend months manually constructing assemblies of ~100 atoms and molecules.
The paper makes two main contributions:
Assembly construction: An algorithm that first computes an optimal assignment of molecules to target positions, then uses SMT constraint solving to find a collision-free execution schedule. When no valid schedule exists, the algorithm identifies conflicting assignments via an unsatisfiable core and iterates with updated distances. The movement of a molecule to its target configuration is learned by RL.
NanoAssemblyGym: A high-fidelity simulation environment built on the Gymnasium API, modeling the stochastic physics of STM manipulation (Gaussian translational noise, discrete rotational responses). RL agents (PPO, TQC, CrossQ) are trained within this environment to execute individual molecule movements while staying within prescribed safety corridors.
Experiments demonstrate assembly of structures up to 420 molecules across three types with near-perfect success rates. Planning completes within minutes even for the largest assemblies, and CrossQ consistently outperforms the other RL algorithms for more complex molecules.

**Compliance With Llm Reviewing Policy:**

Affirmed.

**Final Justification:**

My concerns have been resolved and I keep my original score of accept. I would also like to note that this is an important work that can support future work on this topic. I will also provide an RL based benchmark for future works and the NanoAssemblyGym that lowers the entry barrier for future RL based research.

**Key Questions For Authors:**

1. The Hungarian algorithm guarantees a distance-optimal assignment in the first iteration, but subsequent iterations modify the distance matrix by setting entries to infinity to resolve conflicts. How much does total travel distance increase across iterations in practice?

**Limitations:**

yes.

**Strengths And Weaknesses:**

Soundness
Strengths: The technical approach is rigorous and well-grounded. The two-phase framework is formally specified with precise geometric definitions. RL training is evaluated across multiple algorithms and corridor widths, averaged over 10 runs. The near-zero collision rate across 30,000+ molecule movements is a significant empirical result.
Weaknesses: All experiments are purely simulation-based. While the molecular dynamics are parameterized from real measurements, the sim-to-real gap is unaddressed in the paper and left to future work.

Presentation
Very well-written. The two-phase structure is clearly explained, figures are informative, and the appendix provides thorough reproducibility detail including full algorithms, observation spaces, and training curves. No significant weaknesses.

Significance
The contribution is highly significant within nanostructure formation. Scaling autonomous assembly from ~100 to 420 molecules is an impactful leap. NanoAssemblyGym as a reusable benchmark environment is a significant contribution that will enable future research.

Originality
The combination of SMT-based scheduling with RL execution for nanoscale manipulation is novel and well-motivated. Handling rotational degrees of freedom distinguishes this from prior atom-only work. The framework is a creative and principled integration of constraint solving, combinatorial optimization, and deep RL for a real physical domain.

---

> ### Author Rebuttal · Authors · 2026-03-31
>
> We thank the reviewer for the thorough and thoughtful evaluation, and for recognizing the technical rigor, clarity, and significance of our work. We particularly appreciate the positive assessment of the framework design, experimental results, and the potential impact of NanoAssemblyGym. In the following, we address the sim-to-real gap and respond to your question.
>
> # W1: Missing Sim-to-Real Discussion
> We agree that a discussion of the sim-to-real gap will strengthen the paper, and we will add the following discussion to the paper:
> In order to autonomously control an STM in practice using our simulation environment, the following challenges need to be resolved:
>
> (i) **Changes to the tip**:
> The biggest challenge lies in the fact that the tip may change during operation. In cases where the tip absorbs an atom or molecule, the response to manipulations changes significantly, such that controlled manipulation of the molecule is no longer possible (even with robust RL policies). Recent work [1] addresses the problem of using RL for tip conditioning. As a next step, we will incorporate monitoring mechanisms to detect tip changes into our framework and develop recovery mechanisms.
>
> (ii) **Model Inaccuracies**:
> The responses of molecules to manipulation are inferred from real-world measurements of the molecules’ reactions. Given the vast parameter space, the inferred distributions may be imprecise due to the complex, probabilistic movements and the limited number of available measurements. In future work, we will explore algorithms for training RL policies that ensure safe manipulation under distributional shifts.
> Not affected by the sim-to-real gap:
> NanoAssemblyGym is a high-fidelity simulation environment designed for exact planning at the nanometer scale. Thus, the library performs exact computations of the molecule’s position and orientation on the surface (currently using a lattice constant of L=0.3nm, which can be adjusted by the user depending on the material being modeled), without introducing any abstractions or discretisation in this computation.
>
> [1] Chen, Shiyang, et al. "A Reinforcement Learning-Based Method for Automated Repair of Scanning Probe Tips in a Simulated Environment." 2025 38th International Vacuum Nanoelectronics Conference (IVNC). IEEE, 2025.
>
>
> # Q1: **Changes in Total Travel Distance in Practice**
>
> The changes in total travel distance are negligible, often amounting to only a few nanometers. Across all experiments, the highest deviation observed was 10 nm, which occurred when constructing honeycomb assemblies with an initial optimal travel distance of approximately 8500 nm.

---

> > ### Author Rebuttal · Reviewer_2WPW · 2026-04-03
> >
> > My concerns have been resolved. Ideally, I would like to see comparison with real-world manual experiments to provide benchmarks. But I believe that even without it, the paper makes contribution that can pave the way for future automation and can serve as a benchmark for future works.

---

> > > ### Author Response · Authors · 2026-04-07
> > >
> > > We thank the reviewer for acknowledging our work and are glad to have resolved all questions and concerns. We appreciate their positive assessment of the significance of our work, as well as their view that NanoAssemblyGym can support future research in autonomous nanoscale assembly.

---

### Decision · Program_Chairs · 2026-04-30

**Decision:**

Accept (regular)

**Comment:**

The paper presents a pipeline for large-scale molecular assembly using STM, combining assignment and SMT-based scheduling with RL-based low-level control, along with a new simulation environment (NanoAssemblyGym).
Novelty: Reviewers generally agree that the integration of constraint solving with RL for nanoscale assembly is creative and well-engineered, and the end-to-end pipeline is non-trivial. However, the contribution is primarily at the systems and application level rather than introducing fundamentally new ML ideas. In particular, the role of RL as a low-level controller is not sufficiently justified against simpler alternatives, and the overall novelty lies more in assembling existing components than advancing core methodology.
Evaluation: The experimental results are impressive in scale, demonstrating assembly of up to 420 molecules with high success rates, which is a significant step forward for the domain. The introduction of NanoAssemblyGym is also valuable. However, the evaluation is entirely simulation-based, with no real-world validation or convincing discussion of sim-to-real transfer. Additionally, the lack of meaningful baselines (e.g., non-RL controllers, heuristic planners) makes it difficult to assess the true contribution of individual components. There is also limited analysis of scalability, computational cost, and failure modes.
Overall, this is a strong and well-executed application paper.